# Optimize the Unseen - Fast NeRF Cleanup with Free Space Prior

**Leo Segre**    **Shai Avidan**

Tel Aviv University
leosegre@mail.tau.ac.il    avidan@eng.tau.ac.il
https://leosegre.github.io/optimize-the-unseen/

## Abstract

Neural Radiance Fields (NeRF) have advanced photorealistic novel view synthesis, but their reliance on photometric reconstruction introduces artifacts, commonly known as "floaters". These artifacts degrade novel view quality, particularly in unseen regions where NeRF optimization is unconstrained. We propose a fast, post-hoc NeRF cleanup method that eliminates such artifacts by enforcing a Free Space Prior, ensuring that unseen regions remain empty while preserving the structure of observed areas. Unlike existing approaches that rely on Maximum Likelihood (ML) estimation or complex, data-driven priors, our method adopts a Maximum-a-Posteriori (MAP) approach with a simple yet effective global prior. This enables our method to clean artifacts in both seen and unseen areas, significantly improving novel view quality even in challenging scene regions. Our approach generalizes across diverse NeRF architectures and datasets while requiring **no additional memory** beyond the original NeRF. Compared to state-of-the-art cleanup methods, our method is $2.5\times$ **faster** in inference and completes cleanup training in under 30 seconds.

## 1   Introduction

Neural Radiance Fields (NeRF) have emerged as a leading technique in photorealistic scene reconstruction and novel view synthesis, effectively capturing complex scenes from a limited set of images. However, NeRF's reliance on photometric optimization introduces a significant limitation: it performs poorly in regions of the scene that lack direct observations from the training images. This results in visual artifacts, commonly referred to as "floaters", which are density accumulations in areas unseen by the training cameras. These floaters degrade rendering quality, especially in novel views where the artifacts may obscure scene surfaces or introduce false details, as illustrated in Fig. 1.

While state-of-the-art approaches for NeRF cleanup can mitigate artifacts, they often come with a high computational cost. Nerfbusters [31] uses 3D diffusion models to generate a learned, data-driven 3D prior. This requires NeRF fine-tuning and results in substantial computational demands and extended cleanup training time. BayesRays [10], an alternative approach, reduces cleanup time by performing post-hoc uncertainty-based cleanup; however, it does not modify the NeRF scene itself. Instead, it applies uncertainty-based filtering during inference to clean the rendered frames, which introduces considerable overhead as each frame requires additional uncertainty estimation and processing. These approaches highlight a key trade-off in existing methods between efficient artifact removal and computational expense, underscoring the need for a more effective solution.

Unlike previous methods, which often require architecture-specific modifications or extensive fine-tuning, our approach is robust and generalizes effectively across diverse NeRF architectures and datasets, as demonstrated in Fig. 1. This flexibility makes our method applicable to a wide range of

39th Conference on Neural Information Processing Systems (NeurIPS 2025).

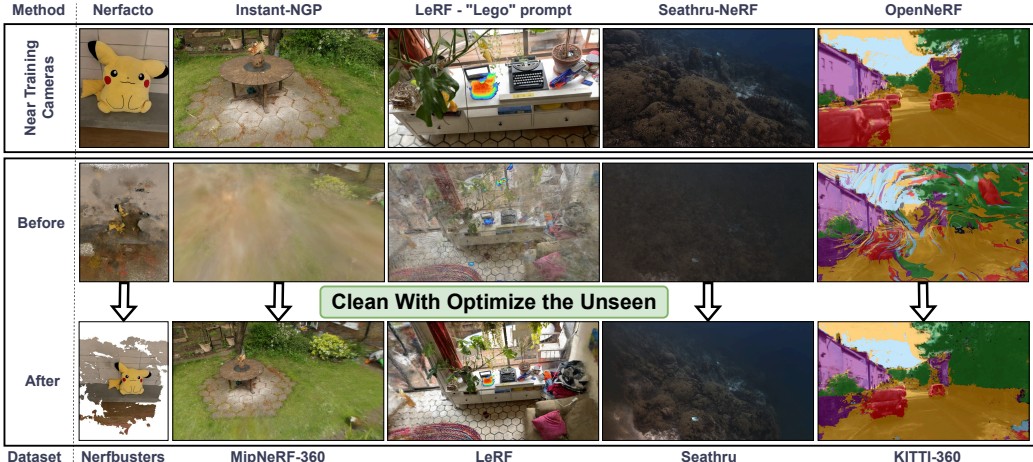

Figure 1: **Density Cleanup Across NeRF Variants and Datasets:** The top row shows novel views taken near the training cameras, while the middle row presents novel views from farther away. The bottom row demonstrates the same distant views after applying our cleanup method, effectively removing artifacts while preserving scene integrity. Our approach generalizes well across a wide range of NeRF methods and datasets - see supplementary material for video demonstrations.

scenarios, outperforming other cleanup methods that are constrained by their reliance on specific priors or additional computational overhead.

A key contribution of our work is that we sample across the entire 3D space, not just along rays. Samples along the rays ensure NeRF fits the given scene, while samples in 3D space introduce a Free Space Prior. This contribution makes our approach lightweight and efficient: we only fine-tune the NeRF itself, without training additional networks or increasing inference-time memory and computation. As a result, floaters are minimized without disrupting the scene's intended structure or introducing additional memory requirements. The method is both efficient and highly compatible with existing NeRF cleanup models, offering a significant improvement in speed and memory efficiency compared to state-of-the-art artifact removal techniques.

The contributions of our method are summarized as follows:

- **Efficient and Lightweight:** Our approach requires *no additional memory* beyond the original NeRF, is *2.5× faster* in inference time, and *1.5× faster* in cleanup training compared to current state-of-the-art methods.

- **Optimization in Unseen Regions:** Unlike previous methods that refine only regions observed by training cameras, our method *actively optimizes density across unseen regions*, enforcing a clean 3D representation and reducing artifacts throughout the scene.

- **Robust and Generalizable:** Our approach *generalizes seamlessly across different NeRF architectures and datasets*, requiring no architectural modifications or learned priors, making it widely applicable to various real-world scenarios.

- **State-of-the-Art Artifact Removal:** Our method effectively eliminates floaters and other density artifacts, *enhancing novel view synthesis quality* while preserving scene details.

## 2 Related Work

**Artifacts in Photometric Optimization** Methods based on photometric reconstruction [21, 20], such as NeRF [22] and 3D Gaussian Splatting [13], have become leading techniques in 3D reconstruction due to their ability to generate high-quality, detailed scene representations by modeling the scene from multi-view images. These methods excel in reconstructing complex geometry and capturing subtle visual details by leveraging the information embedded in photometric data. However, due to their reliance on photometric reconstruction, which inherently contains uncertainties stemming from incomplete or noisy input data, these methods are prone to visual artifacts. While artifacts in

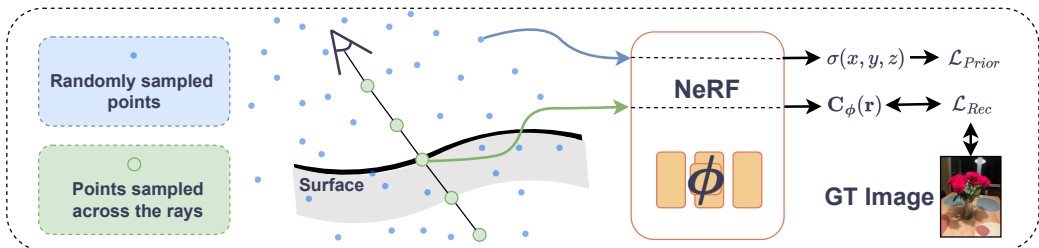

Figure 2: **Overview of Our Cleanup Method:** Our approach fine-tunes a pre-trained NeRF by optimizing density across both seen and unseen regions. We sample points in two ways: (1) along the original training rays (green points), maintaining consistency with the observed scene structure, and (2) randomly across the entire 3D space (blue points), enforcing our Free Space Prior to remove artifacts in unseen areas. Importantly, we also apply the Free Space Prior behind scene surfaces, ensuring that empty regions remain free of unwanted density accumulations. This ensures effective cleanup of floaters while preserving scene integrity.

Gaussian Splatting typically concentrate around scene structures due to imperfect Gaussian optimization [37, 34], NeRF artifacts are more widely distributed across the scene, often impacting novel views severely.

Neural Radiance Fields (NeRFs) [22] often exhibit visual artifacts, such as "floaters" and density inconsistencies, due to training data imperfections, including incomplete scene coverage and data noise. These issues introduce uncertainties in unobserved regions, which can be divided into **aleatoric** and **epistemic** uncertainty. Aleatoric uncertainty, arising from data noise like lighting changes or transient objects, has been addressed in dynamic scenes by works such as NeRF-W [18] and others [26, 25, 11]. In contrast, our focus is on static scenes with epistemic uncertainty, stemming from incomplete scene information. Approaches like CF-NeRF [28] use variational inference to capture this uncertainty in geometry, while BayesRays [10] applies a Bayesian framework to estimate epistemic uncertainty post-training. Our work specifically addresses cleanup of artifacts in static scenes caused by epistemic uncertainty.

**NeRF Cleanup.** Various methods have been proposed to address the visual artifacts common in NeRF reconstructions. Nerfbusters [31] introduced a post-processing technique using diffusion models that learns a local 3D data-driven prior from synthetic data. This data-driven prior, incorporated during NeRF optimization, encourages plausible geometry but requires extensive training and NeRF fine-tuning, adding significant computational overhead. Another approach, BayesRays [10], leverages a post-hoc solution by thresholding an uncertainty field to mask high-uncertainty regions, effectively cleaning artifacts at inference time but at the cost of increased computation during rendering.

These methods, though effective, often come with trade-offs in memory consumption, cleanup time, or inference speed. In contrast, our approach removes artifacts efficiently, preserving the original NeRF's structure and performance without requiring additional models or extensive post-processing, based only on our Free Space Prior. This ensures both speed and practicality in artifact cleanup.

**Prior-Based Optimization** Data-driven optimization is standard in deep learning, but numerous studies demonstrate that incorporating a well-chosen prior can significantly boost performance. For instance, priors have proven effective in inverse problems like image denoising [5, 12, 33, 27, 6, 7]. In particular, [6, 7] introduce a global image prior enforcing sparsity, showcasing how a Bayesian framework can yield a simple yet powerful denoising algorithm. In the context of 3D reconstruction, priors and regularization play similarly critical roles. Nerfbusters [31] uses a local prior to improve scene fidelity and RegNeRF [24] employs a 2D Total Variation (TV) regularizer to rendered images, both maintaining the standard NeRF architecture as we do. Other methods modify the architecture. For example, Plenoxels [35] employs a 3D Total Variation (TV) regularizer to reduce abrupt density changes between neighboring voxels and 2D TV regularization is also applied in factorized plenoptic fields [4, 9]. Plenoctrees [36] incorporates a sparsity loss within a complex pipeline for faster rendering but at high memory and training costs. To address anti-aliasing, Zip-NeRF [3] and MipNeRF-360 [2] applies distortion loss and interlevel loss, reducing artifacts and

improving consistency. While effective, this regularization alone does not eliminate floaters, as our results show, highlighting the need for a stronger constraint on unseen regions.

Inspired by the above, our method employs a global prior to enforce free space within NeRF 3D scenes, enhancing NeRF's rendering performance.

# 3 Background

Understanding NeRF is essential to grasp the significance of our work and its contributions to the field. In this section, we review the fundamental concepts of NeRF and discuss the challenges associated with artifact generation in unobserved regions, which our approach aims to overcome.

## 3.1 Neural Radiance Fields

NeRF [22] is an approach for synthesizing novel views of 3D scenes. Each point in 3D space is represented by a view-dependent radiance and a view-independent density, expressed as follows:

$$\mathbf{c}_\phi(\mathbf{x}, \mathbf{d}), \ \tau_\phi(\mathbf{x}) = \mathcal{R}(\mathbf{x}, \mathbf{d}; \phi) \tag{1}$$

Here, $\phi$ denotes the learnable parameters of the neural network. The color along a ray $\mathbf{r} = \mathbf{o_r} + t \cdot \mathbf{d_r}$ is determined by sampling points $t_i$ along the ray:

$$\mathbf{C}_\phi(\mathbf{r}) = \sum_i \exp\left(-\sum_{j<i} \tau_j \delta_j\right) (1 - \exp(-\tau_i \delta_i)) \, \mathbf{c}_i, \tag{2}$$

In this equation, $\delta_i$ represents the distance between successive sampled points. The optimization of the network parameters $\phi$ aims to minimize the reconstruction loss, which is defined as the squared distance between the predicted and ground truth colors for each ray sampled from the training images $R = \{\mathbf{r}\}_{n=0}^{\mathbf{N}}$.

$$\mathcal{L}_{\text{rec}} = \sum_{\mathbf{r} \in R} \|\mathbf{C}_\phi(\mathbf{r}) - \mathbf{C}_n^{\text{gt}}(\mathbf{r})\|_2^2 \tag{3}$$

An important observation from Eq. (2) is that NeRF's optimization occurs on a per-pixel basis, with the color of each pixel computed along a single ray. Since density decreases exponentially along the ray after it intersects the first surface, there is effectively no optimization beyond that surface, and certainly none where the ray does not pass through. Consequently, unseen regions remain unoptimized, often resulting in unwanted artifacts in these areas.

# 4 Method

Our approach, as shown in Fig. 2, is designed as a post-hoc refinement. It takes a pre-trained NeRF model along with its training cameras and optimize it to remove artifacts in unseen regions while retaining the scene's intended features.

To date, NeRF models, and NeRF cleanup methods in particular, searched for a density field $\sigma$ that best explains the given data. That is, they look for the maximum likelihood solution to the problem. Some methods extend this by learning a general data distribution over multiple 3D scenes, creating a local prior for artifact removal. In contrast, we apply a simple, global prior on the density $\sigma$ - We take the prior $P(\sigma)$ to be the zero density prior.

This global approach not only simplifies the optimization process but also ensures **versatility and robustness** across different NeRF architectures and datasets. Unlike learned priors that depend on specific scene distributions, our method operates effectively regardless of scene complexity or data variations, making it widely applicable and independent of dataset-specific assumptions.

We base our method on the assumption that "There is nothing in the unseen regions". While this assumption does not necessarily hold in real-world scenarios, for NeRF reconstructions, it is more

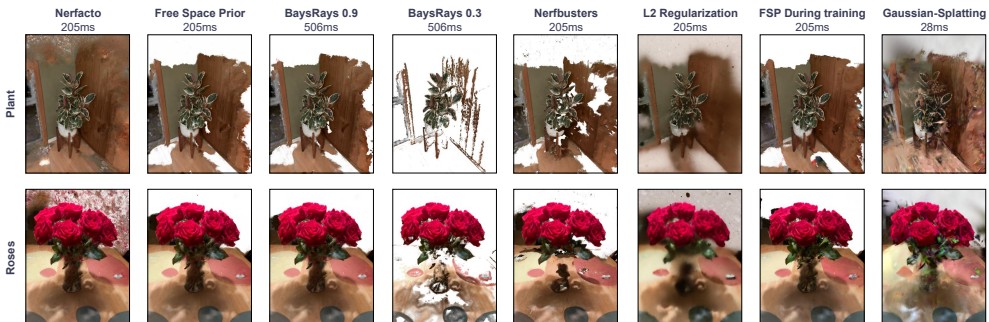

Figure 3: **Qualitative Cleanup Results:** Comparison of cleanup methods on the Plant and Roses scenes from the Nerfbusters dataset, with inference time per frame shown. Each image is a novel view rendered post-cleanup, highlighting the balance between artifact removal and scene coverage. Methods like Free Space Prior and BayesRays (0.9) achieve high coverage with minimal artifacts, while Nerfbusters and BayesRays (0.3) trade coverage for stronger cleanup. Our approach achieves similar results to BayesRays with 40% rendering time.

practical than allowing unseen regions to contain unregulated noise. To enforce this assumption, we apply a sigmoid-softened density prior term that gently drives density $\sigma$ toward zero.

**Enforcing the Free Space Prior**   Ideally, we would like to directly set densities to zero only in the unseen regions. However, determining which regions are strictly unobserved by training rays is challenging due to scene occlusions and structural complexity. Additionally, even if feasible, this direct approach would be computationally expensive. So, instead of explicitly setting densities, we apply the Free Space Prior uniformly across the entire 3D space, affecting both seen and unseen regions. This approach simplifies computation while guiding NeRF optimization to achieve cleaner density distributions.

To implement this, we randomly sample $\mathcal{N}$ points across the 3D space and query the NeRF model $\phi$ for densities at these points, resulting in a set $\Sigma = \{\sigma(\mathbf{x}_i) \mid \mathbf{x}_i \in \mathbb{R}^3, \, i = 1, \ldots, N\}$ of density values. We then construct a softened density term for the Free Space Prior, which we integrate into a Free Space Prior Loss term:

$$\mathcal{L}_{\text{FSP}} = \sum_{i=1}^{N} \left( \frac{1}{1 + e^{-\sigma(\mathbf{x}_i)}} \right)^2 \tag{4}$$

This Free Space Prior Loss optimizes the NeRF parameters $\phi$ by enforcing densities to be zero, reflecting our assumption that "There is nothing in the unseen regions". However, this loss also inadvertently reduces densities in visible regions, where densities should ideally reflect the actual scene structure.

**Balancing Cleanup and Scene Integrity**   Applying the Free Space Prior Loss indiscriminately across 3D space can lead to an empty scene, as the randomly enforced zero-density constraint propagates, causing the MLP to approximate densities as zero throughout the space. To counter this, we introduce a combination of two competing losses: the Free Space Prior Loss Eq. (4) to reduce densities in unseen regions and the NeRF photometric loss Eq. (3) applied along training rays to preserve the scene structure and appearance.

This interaction between losses results in three distinct regions:

- **Unseen Regions:** Only the Free Space Prior Loss affects these regions, pushing densities toward zero and removing floaters. In particular, the densities in occluded regions are also set to zero.

- **Empty Seen Regions (along the ray to the surface):** Both the Free Space Prior Loss and the photometric loss act here, agreeing that the density should be zero.

- **Surface Seen Regions:** Both losses are active, but they compete, as the Free Space Prior Loss encourages lower densities while the photometric loss preserves the actual scene structure.

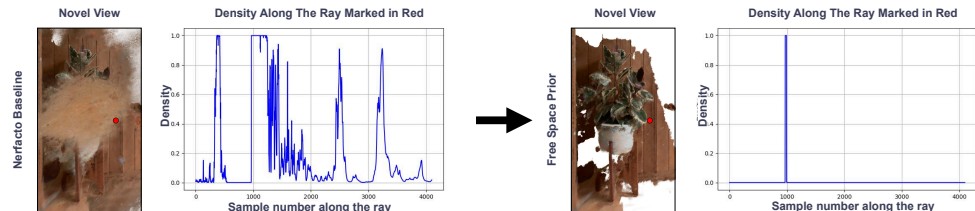

Figure 4: **Density Along a Ray:** (Left) Novel view rendered with the Nerfacto Baseline, with a red dot marking the sampled ray and the density along the marked ray. (Right) The same novel view after our cleanup. Before cleanup, the density along the ray is noisy, with floaters causing unintended peaks. After cleanup, only the surface is captured, and densities in unseen regions are enforced to zero. Densities are sigmoid softened and shifted to [0,1].

Through this balanced combination, empty regions will ultimately reach zero density, eliminating floaters. In contrast, surface regions will settle on a density value that balances both losses, resulting in a reduced yet preserved density that maintains accurate scene reconstruction, as shown in our experiments.

The overall loss function is:

$$\mathcal{L} = \mathcal{L}_{\text{rec}} + \lambda \mathcal{L}_{\text{FSP}} \tag{5}$$

where $\lambda$ controls the trade-off between preserving scene structure and removing artifacts. We set $\lambda = 0.1$ for all visualizations in the paper. $\lambda$ analysis is in the supplementary A.5.

### 4.1 NeRF Regularization

NeRF reconstruction is inherently an ill-posed problem — many solutions can fit the observed images. The highly expressive MLP satisfies the photometric loss along training rays while potentially assigning arbitrary densities in unobserved regions. This results in an under-constrained system where only a subset of solutions is physically meaningful. Drawing a parallel to Ridge regression, we consider $L_2$ regularized MLP to represent the NeRF. The $L_2$ regularizer directly penalizes the parameters, favoring the minimum-norm solution and stabilizing an ill-posed problem.

Yet, there is a difference between the two. $L_2$ regularization places a prior on the weight of the MLP, whereas our Free Space Prior places a prior on the densities themselves. Specifically, instead of regularizing the network weights directly, we regularize a specific component of the output — the predicted densities. Since our objective is formulated as a loss optimized via backpropagation, we explicitly sample the 3D space to enforce this regularization in unobserved regions. This modified approach resolves the ill-posedness by favoring solutions with minimal densities in free space, while preserving the correct photometric reconstruction in observed regions.

## 5 Results

We evaluate our NeRF cleanup method by analyzing its performance in the NeRF cleanup task, comparing both quantitative metrics and *computational efficiency* in terms of cleanup and inference speed. We report numerical results to quantify the reduction of artifacts (floaters) and demonstrate that our approach achieves significant speedup while maintaining high-quality novel view synthesis.

Table 1: **Runtime Comparison:** Inference time (ms/frame) is averaged across all Nerfbusters scenes. Both Nerfbusters and Free Space Prior use only the fine-tuned NeRF at inference, resulting in identical inference runtime.

| Method | Cleanup Time (s) | Inference Time (ms) |
|---|---|---|
| Nerfbusters | 1200 | **205** |
| BayesRays | 37 | 506 |
| Free Space Prior (Ours) | **25** | **205** |

Beyond numerical evaluations, we qualitatively assess our method across a diverse range of NeRF architectures and datasets, showcasing its effectiveness in different scenarios. Additionally, we conduct ablation studies to analyze the impact of key components in our design, offering insights into their role in artifact removal. The implementation details are in the supplementary Appendix A.2.

## 5.1 Dataset and Evaluation Setup

We follow the experimental protocols established by Nerfbusters [31] and BayesRays [10]. Our goal is to assess each method's ability to eliminate these artifacts while preserving the quality and completeness of the reconstructed scenes. For quantitative evaluation, we use the Nerfbusters dataset [31], which is the standard benchmark for NeRF cleanup methods. Unlike other datasets, which primarily focus on reconstructing regions well-covered by training views, Nerfbusters uniquely provides evaluation views that are significantly farther from the training cameras, making it the most challenging and relevant dataset for artifact removal. This ensures that cleanup methods are tested under realistic conditions where unseen regions contribute to artifacts. In addition to the quantitative evaluation on Nerfbusters, we qualitatively evaluate our method across a diverse range of NeRF architectures and datasets, demonstrating its generalizability and effectiveness in different scenarios.

**Coverage Metric**  We adopt the coverage metric as introduced by Nerfbusters [31], which measures the percentage of pixels in the evaluation image that are accurately reconstructed after masking out regions that were either unseen in the training views or are too distant to be relevant (based on a predefined depth threshold).

**Baseline**  Our baseline model (Nerfacto) is built on modern NeRF architectures, incorporating several recent advancements, including hash grid encoding from iNGP [23], proposal sampling and scene contraction from MipNeRF[1], distortion loss from MipNeRF360 [2] as well as per-image appearance optimization from NeRF-in-the-Wild [19]. While these techniques provide a degree of regularization, they fail to prevent the accumulation of artifacts in unseen regions. Our Free Space Prior directly addresses this issue by enforcing explicit density constraints in these areas.

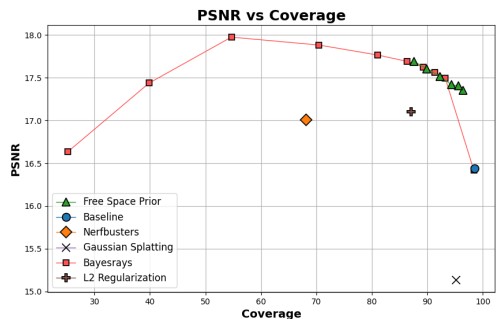

Figure 5: **Quantitative Cleanup Results:** PSNR vs. Coverage comparison of cleanup methods across different thresholds / $\lambda$. Higher values indicate better performance, with the optimal region in the upper-right corner. Results are averaged across all Nerfbusters scenes for a comprehensive evaluation.

To measure performance, we report PSNR and Coverage scores, which quantify both reconstruction quality and the extent to which the scene is accurately rendered without artifacts.

In addition to this standard evaluation (Appendix A.4), we introduce and evaluate a modified experiment that makes the task more realistic and challenging: previous evaluations calculated PSNR, SSIM, and LPIPS using the *ground truth* (GT) mask, which only considers the object region. This can overestimate performance, as artifacts outside the GT mask are ignored. In our setup, the metrics computed over the *predicted* mask, penalizing methods that leave artifacts outside the GT mask (Appendix A.6). This stricter evaluation ensures that methods are assessed on their true ability to remove artifacts while preserving scene integrity.

## 5.2 Performance Comparison

**Cleanup Performance**  We evaluate our approach both qualitatively (Fig. 3) and quantitatively (Fig. 5), using the challenging modified evaluation (i.e., using the predicted mask to evaluate results).

Striking a balance between PSNR and coverage is crucial, as high PSNR with low coverage can be misleading. For instance, as shown in Fig. 5, BayesRays at a 0.3 threshold achieves the highest PSNR but suffers from low coverage, leading to noticeable gaps in the reconstructed scene (Fig. 3) that compromise overall image quality.

As also seen in Fig. 5, Free Space Prior demonstrates high coverage while maintaining a PSNR comparable to BayesRays, the current state-of-the-art in NeRF cleanup. Among the other methods, Nerfbusters exhibits significantly lower coverage and PSNR, indicating excessive artifact removal. In contrast, L2 regularization, Gaussian Splatting, and the Nerfacto baseline exhibit excellent coverage

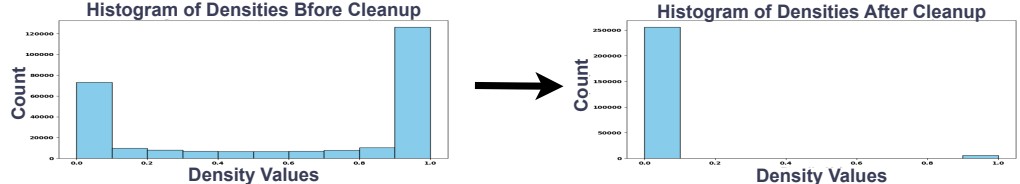

Figure 6: **Density Histogram:** Histogram of densities across the Pikachu scene after sigmoid-softened, shifted to [0, 1]. (Left) Histogram of densities across the scene before the cleanup process. (Right) Histogram of the densities across the scene after the cleanup process.

but lower PSNR, retaining more scene information while struggling with artifact suppression and overall scene fidelity, highlighting the dataset's challenge.

Our method, as well as BayesRays with a 0.9 threshold, produce visually superior results that retain the scene's structure and coverage, making them more visually appealing and accurate to a human eye. These images highlight that our method preserves both high coverage and high PSNR, outperforming Nerfbusters and L2 regularization, which, while comparable in inference time, achieves lower PSNR and coverage than our method.

For completeness, we show that our method also works if applied during the training phase when learning the NeRF from scratch. In that case, our method makes the cleanup phase redundant. The visual results ($\lambda = 10^{-5}$) are shown in Fig. 3 and the numerical details are in the appendix Tab. 4.

**Runtime Performance**  Our method combines fast cleanup and inference times (Tab. 1) while maintaining high-quality artifact cleanup (Fig. 5). For inference, both Nerfbusters and our method achieve a per-frame rendering time of 205 ms, as each directly fine-tunes the NeRF without adding additional overhead at inference. In contrast, BayesRays incurs a significant slowdown, with a per-frame inference time of 506 ms—2.5 times slower—due to its additional uncertainty thresholding applied during rendering.

For cleanup time, Nerfbusters employs a pre-trained 3D diffusion model to fine-tune NeRF per scene, resulting in a high cleanup time of approximately 20 minutes per scene. BayesRays is faster, taking 37 seconds to gather data for Hessian computation, while our approach is the fastest, completing cleanup training in just 25 seconds.

## 5.3  Geometric Evaluation

While NeRF is a photometric based method, it represents a 3D scene with geometric properties. To evaluate the cleanup quality, in addition to the photometric evaluation we conducted three geometric experiments on the Nerfbusters dataset as shown in Table 2.

Table 2: Comparison of methods on geometric metrics.

| Method | Chamfer Distance↓ | Depth (RMSE)↓ | Depth (MAE)↓ | PSNR↑ |
|---|---|---|---|---|
| Ours () | **0.011** | **3.80** | **1.02** | **28.64** |
| BayesRays (T=0.9) | **0.011** | 3.91 | 1.07 | 28.51 |

**Chamfer Distance:** We measured Chamfer Distance between pre-cleanup and post-cleanup scenes. For each scene, we generated 35k-point point clouds using training cameras and NeRF reconstructions (pre- and post-cleanup), then computed Chamfer Distance between them.

**Depth Consistency:** We evaluated surface preservation using depth maps to check whether they remain consistent after cleanup on the training images. For the most accurate measurement, we compared rendered depth maps to pseudo-gt depth maps derived from a NeRF trained on both training and test images.

**PSNR on Training Images:** We measured the PSNR of training rendered images after cleanup. The rationale is that validating scene integrity using training images ensures that the core mechanism of NeRF reconstruction is preserved even through the cleanup phase.

All three geometric evaluations confirm that our uniform sampling strategy preserves geometric details effectively as our method achieves comparable or better scores compared to BayesRays.

## 5.4 Robustness

Our method is designed to be robust across different NeRF architectures and datasets, ensuring effective artifact removal across a variety of conditions. Unlike methods that rely on architecture-specific modifications or learned priors trained on specific datasets, our Free Space Prior provides a generalizable solution that applies seamlessly to different NeRF formulations.

As shown in Fig. 1, we evaluate our method across diverse NeRF architectures, including Nerfacto [30], Instant-NGP [23], LeRF [14], Seathru-NeRF [16], and Open-NeRF [8], each of which has distinct optimization techniques and applications. Our approach also generalizes well across challenging datasets, such as Nerfbusters [31], MipNeRF-360 [2], LeRF [14], Seathru [16], and KITTI-360 [17], covering a range of real-world and synthetic environments. These results highlight the versatility of our method, effectively reducing artifacts without requiring architecture-specific tuning.

Our prior works on the Seathru-NeRF dataset, which operates in extreme underwater conditions. The physics of underwater imaging involves attenuation and back scatter, so density should not be zero. Yet, our prior reduces artifacts significantly. See Fig. 1, column 4. Beyond novel view synthesis, our cleanup process enhances downstream tasks such as NeRF-based semantic segmentation and language-based scene understanding, as shown in Fig. 1, columns 3 and 5.

## 5.5 Method Analysis

To better understand the effect of our Free Space Prior, we analyze how it modifies the density distribution within the NeRF.

**Density Histogram Analysis**   We visualize the density distribution before and after cleanup by sampling points randomly across the 3D space and plotting their density values, as shown in Fig. 6. Before cleanup, the density histogram exhibits a wide distribution, with intermediate densities appearing throughout the scene. These intermediate values correspond to unwanted density accumulations (floaters) in unseen regions, which lack sufficient supervision during NeRF optimization.

After applying our Free Space Prior, the density distribution shifts toward a bimodal structure, where densities are either near zero (free space) or high (scene surfaces). This indicates that our method successfully removes floaters while preserving scene structures, leading to a cleaner and more physically meaningful density field. See Appendix A.9 for underwater analysis.

**Density Along a Ray**   To further illustrate the effect of our cleanup, we examine the density values along a sampled ray, as shown in Fig. 4. Before cleanup, the ray passes through multiple unintended density peaks, corresponding to floating artifacts that degrade novel view synthesis. After cleanup, these spurious densities are eliminated, and the ray exhibits only a single, well-defined peak at the true surface, confirming that our method effectively enforces physically consistent density distributions.

## 5.6 Ablation Study

**The Number of Randomly Sampled Points**
The rational behind selecting $N$ is that $N$ should be proportional to the reconstruction loss sampling density. In each iteration, we use 4,096 rays with 352 samples per ray ( 1.45M samples) for reconstruction loss. We chose ( 131K samples), which is approximately one order of magnitude smaller, ensuring the FSP loss provides a meaningful cleanup signal while adding minimal computational overhead.

The ablation study in Fig. 7 confirms our rationale, $N = 2^{17}$ achieves substantial cleanup improvements with only 10% additional runtime compared to minimal sampling, while avoiding the diminishing returns of higher values.

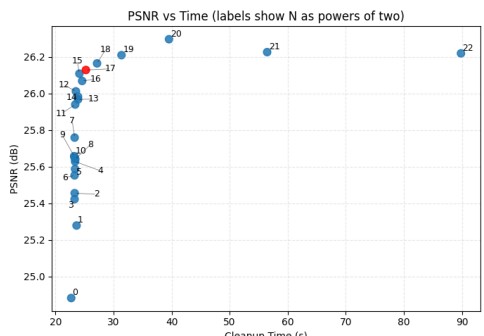

Figure 7: **Number of Samples Study:** PSNR vs. Cleanup Time of the Pikachu scene for different $N$ values. Our choice of $N = 2^{17}$ marked in red.

Table 3: Comparison of different methods under sparse and dense view settings.

| # Train Images | Sparse View | | | | | | | | | | | | Dense View | | |
| | 24 images (Every 8th) | | | 47 images (Every 4th) | | | 93 images (Every 2nd) | | | 161 images (Regular) | | | 8 Out of 9 | | |
| Method | PSNR↑ | SSIM↑ | LPIPS↓ | PSNR↑ | SSIM↑ | LPIPS↓ | PSNR↑ | SSIM↑ | LPIPS↓ | PSNR↑ | SSIM↑ | LPIPS↓ | PSNR↑ | SSIM↑ | LPIPS↓ |
|---|---|---|---|---|---|---|---|---|---|---|---|---|---|---|---|
| Base | 22.32 | 0.637 | 0.261 | 24.41 | 0.703 | 0.226 | 25.33 | 0.724 | 0.212 | 25.74 | 0.731 | 0.212 | 27.74 | 0.789 | 0.181 |
| BayesRays | **22.33** | 0.637 | 0.259 | **24.42** | 0.703 | 0.226 | **25.38** | 0.725 | 0.212 | **25.74** | 0.731 | **0.212** | **27.73** | **0.789** | **0.181** |
| Ours | 22.31 | **0.640** | **0.257** | 24.40 | **0.705** | **0.222** | 25.35 | **0.727** | **0.211** | 25.74 | **0.732** | 0.213 | 27.70 | 0.788 | 0.183 |

**Sparse and Dense View Performance**  To overcome concerns around performance degradation we conducted an ablation study with sparse and dense view settings, comparing our method with $\lambda = 0.01$ and BayesRays with $T = 0.9$, which achieve comparable coverage rates. The sparse view study is conducted on the Garden scene from mipNeRF-360 dataset. The results on Table 3 demonstrate that our method not only preserves NeRF's interpolation capabilities under sparse training conditions but actually improves perceptual quality (SSIM/LPIPS) while maintaining comparable PSNR.

The dense view study is conducted on the mipNeRF-360 dataset and as shown in Table 3 our method has a negligible influence and it maintains high-quality results while being 2.5× faster in inference compared to BayesRays.

**Forcing Prior Along a Ray**  A key design choice in our method is sampling density across the entire 3D space. We conduct an ablation study to *assess the impact of this global sampling strategy* on artifact removal.

An alternative approach would be to apply the prior only to points sampled along training rays, as commonly done in NeRF-based regularization methods. We compare our full 3D sampling strategy with one that applies the prior loss solely along training rays, varying the number of samples per ray. As shown in Fig. 8, limiting optimization to training rays results in incomplete cleanup and leaves floaters in unseen regions.

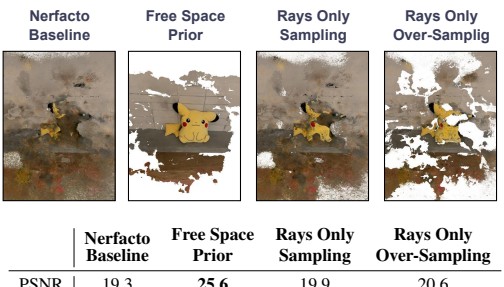

| | Nerfacto Baseline | Free Space Prior | Rays Only Sampling | Rays Only Over-Sampling |
|---|---|---|---|---|
| PSNR | 19.3 | **25.6** | 19.9 | 20.6 |

Figure 8: **Enforcing the Prior Along Training Rays:** (Top) cleanup result. (Bottom) PSNR on Pikachu scene.

Even with more samples per ray, cleanup improves only slightly and does not match the performance of full 3D sampling, which better regularizes completely unseen regions.

These results confirm that sampling beyond training rays is essential for effective NeRF cleanup. By enforcing the Free Space Prior across the entire 3D scene, our method ensures that unseen regions remain artifact-free, leading to significantly improved novel view quality.

# 6 Limitations

The method optimizes unseen regions while using reconstruction loss to preserve the original scene. As a result, it cannot correct artifacts caused by image inconsistencies, moving objects, poor bundle adjustment, or noisy input images. It is only effective for artifacts resulting from under-optimization in specific regions of the scene. An example of the limitation using noisy images is in Appendix A.8.

# 7 Conclusion

We present an efficient cleanup method for NeRFs that leverages a simple yet effective Free Space Prior to significantly reduce artifacts, particularly floaters, in unseen regions. Our approach eliminates these artifacts without modifying the NeRF architecture or increasing memory overhead, making it a lightweight and scalable solution. We demonstrate that our method outperforms SOTA cleanup approaches in both efficiency and quality, achieving up to $2.5\times$ faster inference, $1.5\times$ faster cleanup, and requiring no additional memory. Furthermore, our approach is robust across diverse NeRF architectures and datasets, effectively reducing artifacts in a variety of real-world and synthetic scenarios. By preserving scene fidelity while ensuring consistent and generalizable artifact removal, our method provides a practical and widely applicable solution for enhancing NeRF-based NVS.

## Acknowledgments

Parts of the research were supported by an Israel Innovation Authority Grant.

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

# A   Technical Appendices and Supplementary Material

## A.1   Appendix Overview

This document contains supplemental material regarding the following topics:

- **A.1 Implementation Details**
- **A.2 Appendix Overview**
- **A.3 Method Robustness** - Discussion and visualization of the robustness of our method.
- **A.4 Evaluation Using the Old Paradigm** - Evaluation using the original Nerfbusters evaluation.
- **A.5 Choosing $\lambda$** - Discussion about the temperature parameter $\lambda$
- **A.6 Full Quantitative Results** - The numerical results including a discussion about Dice score as an alternative to coverage metric.
- **A.7 Additional Qualitative Results** - more visual examples of our method.
- **A.8 Noisy images Study** - Study about the limitations.
- **A.9 Densities in Complex Medium** - Underwater analysis.

## A.2   Implementation Details

All experiments were conducted using the latest version of Nerfacto and Splatfacto from Nerfstudio [30], pre-trained for 30,000 steps. Comparisons between different methods were performed on the same pre-trained base NeRF model to ensure consistency. In our method, density optimization was achieved with 1,000 iterations using $2^{17}$ randomly sampled points across the 3D scene space. Optimization was carried out through Nerfacto's optimizers. All experiments were run on a single NVIDIA RTX A5000 GPU. Results for BayesRays [10] and Nerfbusters [31] were generated using their respective pipelines, including the pre-trained 3D diffusion model weights provided by Nerfbusters. The predicted mask indicates the presence of "something" in a pixel after cleanup, created by thresholding the density accumulation at 0.98. The splatfacto model is based on Gaussian Splatting [13] with regularization to prevent artifacts [34, 32].

## A.3   Method Robustness

The Nerfbusters dataset [31] is specifically designed to test the cleanup task with challenging evaluation camera positions, making it an excellent benchmark for assessing artifact removal. However, the floater artifact phenomenon is not unique to Nerfbusters—it is a common issue across NeRF scenes. The primary limitation of other datasets is the lack of challenging evaluation cameras that can serve as a reliable ground truth for cleanup assessment.

To demonstrate the robustness and generalizability of our method, we evaluate it qualitatively on scenes from other widely used datasets. Specifically, we include results for the Garden scene from the mip-NeRF 360 dataset [2], the Basket scene from the Light Fields (LF) dataset [38], the Playground scene the from Tanks and Temples dataset [15], and the Trevi scene from the PhotoTourism dataset [29]. These scenes provide diverse settings with varying levels of complexity, testing the adaptability of our approach.

In Fig. 9, and the attached videos, we compare our method to the Nerfacto baseline. The results confirm that our method effectively removes artifacts while preserving scene details across a wide range of scenes, underscoring its robustness beyond the Nerfbusters dataset.

## A.4   Evaluation Using the Old Paradigm

As stated in the main paper, the conventional metric comparison has relied on GT masks to measure PSNR and Coverage. This approach allows methods that do not clean regions outside the area of interest to achieve high PSNR and, more importantly, artificially inflated Coverage. In Fig. 10, we demonstrate that this old paradigm introduces a significant bias in favor of the baseline, despite its practical noisiness, as it disregards artifact-heavy regions outside the GT masks. Moreover, we

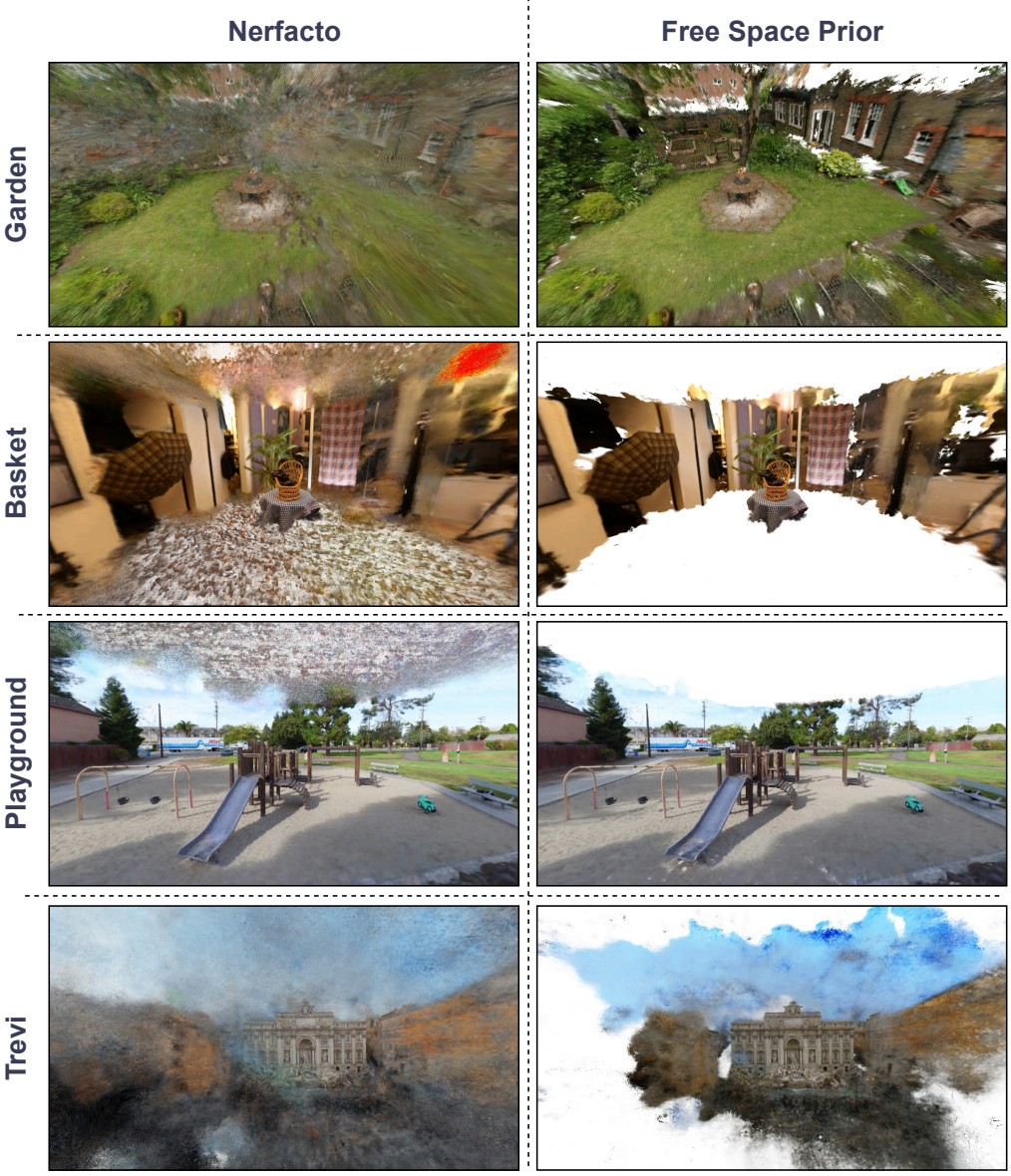

Figure 9: **Qualitative comparison of cleanup results on various scenes:** (Left) Novel view rendered using the Nerfacto model. (Right) The same view after applying our Free Space Prior cleanup. The evaluated scenes include **Garden** from the mip-NeRF 360 dataset, **Basket** from the Light Fields (LF) dataset, **Playground** from the Tanks and Temples dataset, and **Trevi** from the PhotoTourism dataset. Our method effectively removes artifacts while preserving scene details across diverse datasets.

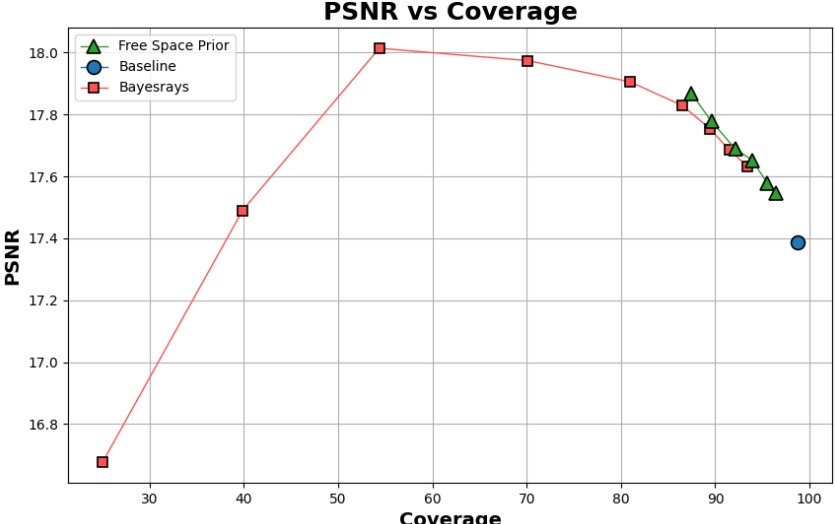

Figure 10: **Evaluation using old paradigm:** Comparison of our method, BayesRays, and the baseline on the Nerfbusters dataset, calculating PSNR solely within the GT masks.

show that our method achieves SOTA performance even under the conventional paradigm, while also excelling in our upgraded metric paradigm.

## A.5 Choosing $\lambda$

Our method is robust to a wide range of $\lambda$ values as can be seen in 4, $\lambda \in [10^{-5}, 1]$ yields both high PSNR (17.35–17.69) and high coverage (87.57%–96.45%). This robustness stems from our method's design, which optimizes regions where the reconstruction loss is inactive. Even with small $\lambda$ values, the FSP loss dominates in these regions, yielding much cleaner results than default Nerfacto. We did not include experiments with $\lambda > 1$, as this would imbalance the losses and violate the principle that the FSP loss should complement, not compete with, the reconstruction loss.

## A.6 Full Quantitative Results

Tab. 4 presents the complete evaluation results for all methods, directly correlating with the points depicted in Figure 4 of the main paper. Each row in Tab. 4 corresponds to a point in the PSNR vs. Coverage plot. Beyond the standard PSNR and Coverage metrics shown in the main paper, we also report SSIM, LPIPS, and Dice scores for a more comprehensive evaluation. While SSIM and LPIPS have been previously evaluated for BayesRays, we introduce the Dice score to provide additional insights into the cleanup results.

**Dice Score** To complement the coverage metric, we propose the Dice score as a meaningful addition for evaluating NeRF cleanup. The Dice score, widely used in segmentation tasks, quantifies the similarity between two sets by measuring their overlap. In the context of NeRF cleanup, it reflects how effectively a method removes unwanted artifacts. Specifically, it assesses how well a method distinguishes between the intended reconstruction and regions that should remain empty.

The Dice score offers a nuanced perspective that the coverage metric alone cannot provide. Coverage measures how completely a method reconstructs regions visible from the training cameras but does not penalize excessive retention of artifacts. In contrast, the Dice score penalizes methods that inaccurately retain content in unseen areas. For example, as shown in Tab. 4, the baseline achieves a high coverage score because it does not perform any cleanup. However, Tab. 4 also reveals a low Dice score for the baseline, highlighting its inability to remove artifacts from unseen regions effectively.

This additional metric reinforces the importance of balancing reconstruction accuracy with artifact removal to achieve high-quality NeRF cleanups.

Table 4: **Full Results:** Comprehensive evaluation metrics for the methods discussed in our paper. Each row corresponds to a point in the PSNR vs. Coverage graph (Figure 4) from the main paper, providing additional insights into the trade-offs between reconstruction accuracy and artifact removal.

| Method | Threshold / $\lambda$ | PSNR | SSIM | LPIPS | Coverage (%) | Dice |
|---|---|---|---|---|---|---|
| Nerfacto (Baseline) | N/A | 16.44 | 0.53 | 0.45 | 98.48 | 0.83 |
| Ours | $10^0$ | 17.69 | 0.62 | 0.30 | 87.57 | 0.87 |
|  | $10^{-1}$ | 17.60 | 0.60 | 0.32 | 89.88 | 0.88 |
|  | $10^{-2}$ | 17.52 | 0.58 | 0.35 | 92.26 | 0.89 |
|  | $10^{-3}$ | 17.42 | 0.57 | 0.37 | 94.35 | 0.90 |
|  | $10^{-4}$ | 17.40 | 0.56 | 0.39 | 95.60 | 0.90 |
|  | $10^{-5}$ | 17.35 | 0.56 | 0.40 | 96.45 | 0.90 |
| Nerfbusters | N/A | 17.01 | 0.64 | 0.27 | 68.13 | 0.73 |
| BayesRays | 0.1 | 16.64 | 0.70 | 0.19 | 25.21 | 0.43 |
|  | 0.2 | 17.44 | 0.68 | 0.20 | 39.86 | 0.54 |
|  | 0.3 | 17.97 | 0.67 | 0.23 | 54.67 | 0.65 |
|  | 0.4 | 17.88 | 0.65 | 0.26 | 70.44 | 0.76 |
|  | 0.5 | 17.77 | 0.63 | 0.28 | 81.04 | 0.83 |
|  | 0.6 | 17.69 | 0.62 | 0.30 | 86.36 | 0.87 |
|  | 0.7 | 17.63 | 0.61 | 0.32 | 89.27 | 0.88 |
|  | 0.8 | 17.56 | 0.59 | 0.34 | 91.29 | 0.89 |
|  | 0.9 | 17.49 | 0.58 | 0.36 | 93.23 | 0.90 |
|  | 1.0 | 16.42 | 0.53 | 0.45 | 98.36 | 0.82 |
| Gaussian Splatting | N/A | 15.13 | 0.50 | 0.41 | 95.22 | 0.89 |
| L2 Regularization | N/A | 17.10 | 0.54 | 0.52 | 87.09 | 0.87 |
| Ours - During Training Phase | $10^0$ | 14.11 | 0.48 | 0.48 | 78.97 | 0.73 |
|  | $10^{-1}$ | 16.35 | 0.56 | 0.38 | 84.16 | 0.85 |
|  | $10^{-2}$ | 17.16 | 0.58 | 0.36 | 89.51 | 0.88 |
|  | $10^{-3}$ | 17.43 | 0.58 | 0.36 | 92.42 | 0.90 |
|  | $10^{-4}$ | 17.51 | 0.57 | 0.37 | 93.38 | 0.90 |
|  | $10^{-5}$ | 17.30 | 0.56 | 0.39 | 95.21 | 0.91 |

## A.7 Additional Qualitative Results

We present qualitative results on additional scenes from the Nerfbusters dataset in Fig. 11. These examples support our claim of achieving a balance between effective cleanup and scene preservation.

Furthermore, the qualitative results highlight the limitations of Gaussian Splatting. Notably, it exhibits significant noise in novel views located far from the training cameras. This issue aligns with the quantitative findings, as the excessive noise contributes to the low PSNR scores reported in Tab. 4. These observations underline the importance of robust artifact removal in achieving high-quality reconstructions for novel viewpoints.

In addition to the static images, we attach videos of the Pikachu, Car, and Plant scenes from the Nerfbusters dataset. These videos provide side-by-side comparisons of the Nerfacto baseline and our Free Space Prior results. The comparisons clearly illustrate that our method delivers consistent artifact removal across the entire scene, rather than affecting only isolated parts visible in the picked images. This consistency highlights our method's robustness in addressing both local and global cleanup challenges while preserving the scene's overall structure and quality.

## A.8 Noisy Images Study

As stated in our limitations section, our approach is based on the reconstruction loss, which relays on a good data. To demonstrate this, we added gaussian noise to the input images of a scene, build a NeRF and applied our method on the noisy scene. As can be seen in Fig. 13, the cleanup did work on under-optimized regions, but did not fix the color bias caused by the noisy input images.

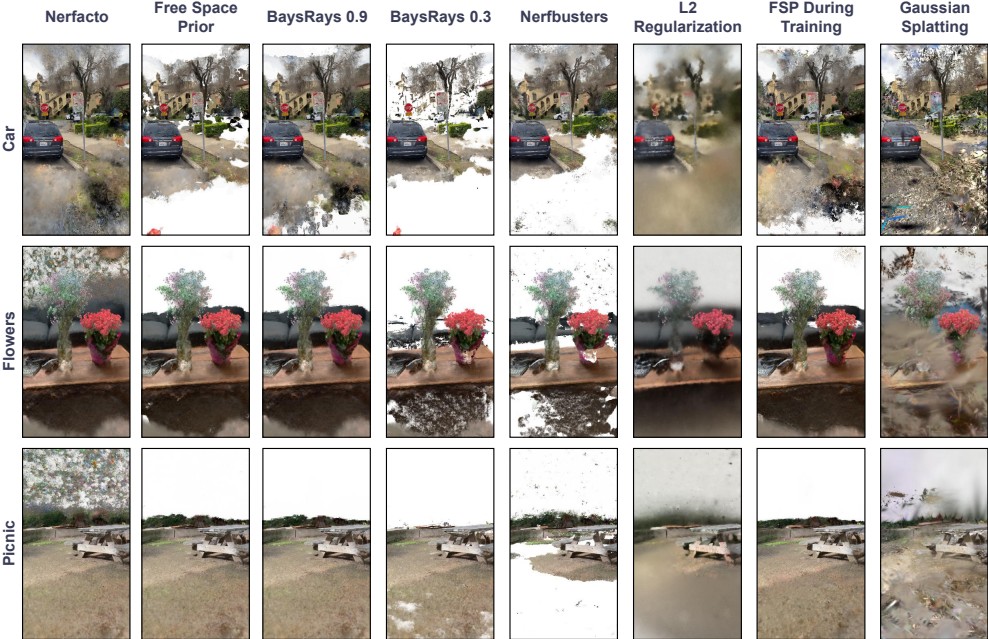

Figure 11: **Qualitative Cleanup Results:** Visualization of different cleanup methods applied to the Car, Flowers and Picnic scenes from the Nerfbusters dataset. Each image shows a novel view from the evaluation set rendered post-cleanup.

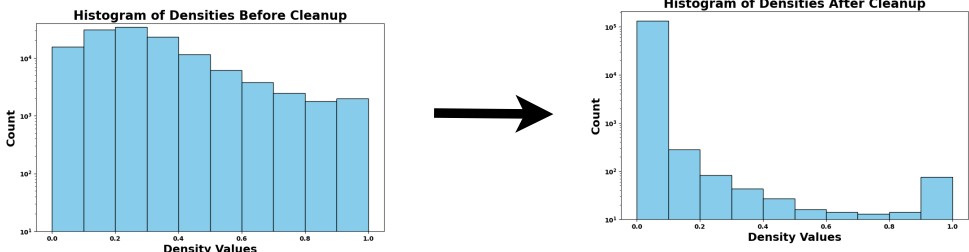

Figure 12: **Density Histogram Underwater:** Histogram of densities across the Curacao scene after sigmoid-softened, shifted to [0, 1]. (Left) Histogram of densities across the scene before the cleanup process. (Right) Histogram of the densities across the scene after the cleanup process.

## A.9 Densities in Complex Medium

As seen in Fig. 1, our approach works on underwater scenes like Seathru dataset. But the assumption that "There is nothing in the unseen area" does not mean densities should be zero. This is because the physics of underwater imaging involve attenuation and back scattering, which means density should not be zero.

We investigate this further by calculating a density histogram for the underwater scene Curacao from the Seathru dataset. See Fig. 12. As can be seen, the original density histogram (left) is pushed towards the bi-modal histogram (right) that we find in regular images (cf. see Fig. 6). Because the prior does not match the imaging physics the resulting histogram is not bi-modal, leaving enough non-zero densities to handle the image properly, as shown visually by the results on the Seathru dataset.

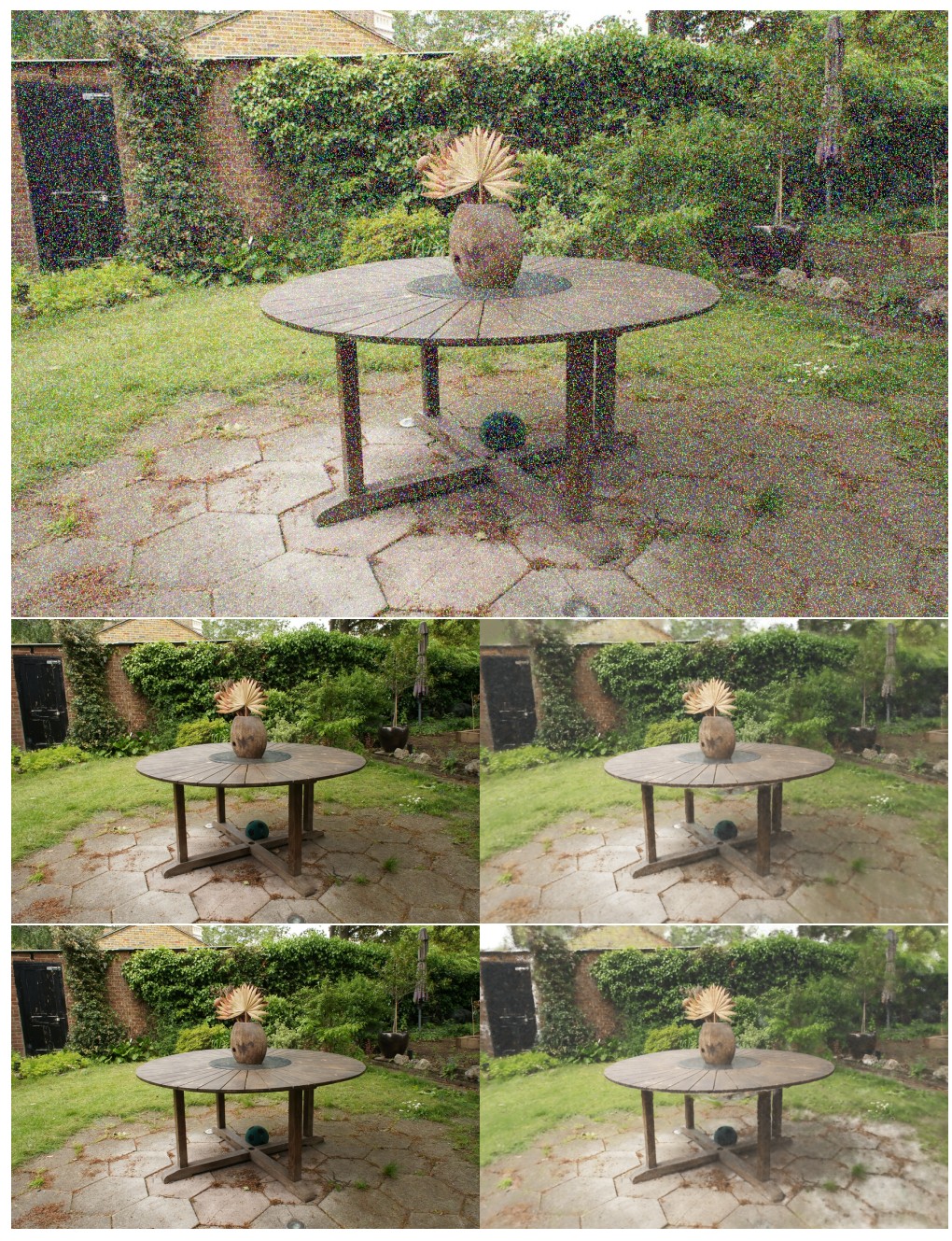

Figure 13: **Noisy input images study:** (Top) Noised input image. (Middle) Original images and the novel view from the Nerfacto baseline. (Bottom) Original images and the novel view after cleanup.

