# OpenReview forum: "Optimize the Unseen - Fast NeRF Cleanup with Free Space Prior"
_NeurIPS.cc/2025/Conference — NeurIPS 2025 poster_

### Official Review · Reviewer_pWFp · 2025-06-01

**Clarity:** 3
**Significance:** 3
**Originality:** 3
**Rating:** 4
**Confidence:** 4

**Summary:**

This paper introduces a Free-Space Prior as a fast, post-hoc method to eliminate floaters in NeRF by enforcing zero density in unseen regions through regularization. The approach improves efficiency without adding significant training or memory overhead and generalizes well across architectures and datasets.

**Questions:**

1) What causes the Free-Space Prior to yield worse SSIM and LPIPS values, indicating lower performance on perceptual metrics?
2) How does the Free-Space Prior affect fine structure reconstruction? Does it impact surface reconstruction accuracy?
3) What is the best approach for selecting hyperparameters across different scenes?

**Ethical Concerns:**

["NO or VERY MINOR ethics concerns only"]

**Final Justification:**

I’m also hesitant between a 3 and a 4. During the rebuttal, the authors effectively demonstrated that their free-space loss has a marginal—almost negligible—impact on fine-grained geometry or surface reconstruction while successfully eliminating floaters. While the method does not outperform BayesRay, I value its simplicity and the potential inspiration it offers for applying 3DGS or NeRF in real-world scenarios. Additionally, I would suggest that refining the sampling strategy may further boost its performance and enhance its overall appeal.

**Limitations:**

yes

**Paper Formatting Concerns:**

No Concern

**Quality:**

2

**Strengths And Weaknesses:**

**Strengths:**

- The method is simple and has the potential to be applied to various NeRF architectures without additional computational overhead.

**Weaknesses:**

- While PSNR results are promising, SSIM and LPIPS metrics lag significantly behind baseline methods.

- The uniform sampling in the free-space loss is overly aggressive, often treating all regions equally and potentially erasing fine details, especially near object boundaries or thin structures. This can lead to loss of important scene content and over-smoothing, particularly in scenes with intricate geometry. The paper does not analyze the extent of this effect or suggest ways to mitigate it.

- The new loss function is highly dependent on careful tuning of the weighting factor \lambda, which determines the balance between artifact removal and scene fidelity. Without detailed analysis or tuning guidelines, the method may not generalize well, and users may struggle to achieve good results out of the box. This lack of robustness and practical instructions undermines the usability of the approach for varied datasets and scene types.  The proposed method also demonstrates a smaller trade-off region. As shown in Figure 5; in comparison, BayesRay offers a wider trade-off region, providing greater flexibility.

---

> ### Author Rebuttal · Authors · 2025-07-29
>
> We thank the reviewer for the feedback, we encourage the reviewer to view the **HTML visualization video** in the supplementary as one of the reviewers mentioned it as a helpful tool to understand the good results. Below are our clarifications and answers:
>
> ## SSIM and LPIPS Performance
>
> We appreciate the reviewer's attention to the SSIM and LPIPS metrics. We understand the confusion may arise from comparing methods at different coverage levels. We would like to clarify that **our method achieves comparable performance to baseline methods** when compared at equivalent coverage levels. We will enhance this clarification in the paper to make the comparison clearer.
>
> **Our method achieves virtually identical SSIM and LPIPS performance to BayesRays at equivalent coverage levels, while being 2.5× faster in inference.**
>
> When comparing at equivalent coverage levels (refer to Table 2 in supplemental - Full Results):
>
> **Coverage ~92%-93%:**
>
> | Method | Threshold / $\\lambda$ | PSNR | SSIM | LPIPS | Coverage (%) | Dice |
> |--------|------------------------|------|------|-------|--------------|------|
> | Ours | 0.01 | 17.52 | 0.58 | 0.35 | 92.26 | 0.89 |
> | BayesRays | 0.9 | 17.49 | 0.58 | 0.36 | 93.23 | 0.9 |
>
> **Coverage ~89%-90%:**
>
> | Method | Threshold / $\\lambda$ | PSNR | SSIM | LPIPS | Coverage (%) | Dice |
> |--------|------------------------|------|------|-------|--------------|------|
> | Ours | 0.1 | 17.60 | 0.60 | 0.32 | 89.88 | 0.88 |
> | BayesRays | 0.7 | 17.63 | 0.61 | 0.32 | 89.27 | 0.88 |
>
>
> ## Uniform Sampling Effectiveness
>
> We respectfully disagree with the assessment that uniform sampling is overly aggressive. Our results demonstrate that we achieve state-of-the-art performance in PSNR, SSIM, and LPIPS while maintaining high coverage. Both qualitative and quantitative results support the effectiveness of our approach without evidence of over-smoothing or detail loss.
>
> ## The Regularization Term (Choosing $\\lambda$)
>
> We would like to address the concern about $\\lambda$ tuning. Our method demonstrates robustness across hyperparameter choices rather than requiring careful tuning. All visualizations in the paper were generated with $\\lambda=0.1$ across different NeRF methods and scene characteristics, demonstrating universal applicability.
>
> Our extensive ablation study (Appendix A.5) shows that our method's mechanism is more important than precise $\\lambda$ tuning:
>
> > *"Our method is robust to a wide range of $\\lambda$ values as can be seen in Table 2, $\\lambda \\in [10^{-5}, 1]$ yields both high PSNR (17.35–17.69) and high coverage (87.57%–96.45%). This robustness stems from our method's design, which optimizes regions where the reconstruction loss is inactive."*
>
> Our "**Balancing Cleanup and Scene Integrity**" section (line 157) explains the 3 regions of interaction between the losses:
>
> - **Unseen regions**: Only FSP loss acts, pushing densities to zero
> - **Empty seen regions**: Both losses agree on zero density
> - **Surface regions**: NeRF reconstruction rays provide strong, consistent supervision by repeatedly sampling the same locations, while the sparse FSP loss only occasionally samples these areas and therefore cannot overpower the reconstruction signal
>
> You can observe Figure 4 to see how density in empty regions is cleaned while preserving surface structure.
>
> ## Trade-off Analysis
>
> While BayesRays offers a wider PSNR-Coverage range, this can be misleading. As shown in Figure 5 and Figure 3 and discussed in lines 250-253:
>
> > *"Striking a balance between PSNR and coverage is crucial, as high PSNR with low coverage can be misleading. For instance, as shown in Fig. 5, BayesRays at a 0.3 threshold achieves the highest PSNR but suffers from low coverage, leading to noticeable gaps in the reconstructed scene (Fig. 3) that compromise overall image quality."*
>
> Our method maintains both high PSNR and coverage in the **practical operating range**.
>
> ## Responses to Specific Questions
>
> **Q: What causes the Free-Space Prior to yield worse SSIM and LPIPS values, indicating lower performance on perceptual metrics?**
>
> A: As demonstrated above, our method does not yield worse values when compared at equivalent coverage levels.
>
> **Q: How does the Free-Space Prior affect fine structure reconstruction? Does it impact surface reconstruction accuracy?**
>
> A: Our results indicate no negative impact on fine structure reconstruction. Figure 4 demonstrates that wall density is well-preserved, and surface reconstruction remains accurate. Both qualitative and quantitative results support maintained reconstruction quality.
>
> **Q: What is the best approach for selecting hyperparameters across different scenes?**
>
> A: We recommend using $\\lambda=0.1$ universally. Our robustness analysis (Appendix A.5) shows this works well across diverse scenes, with the method's core mechanism being more important than precise tuning.

---

> > ### Comment · Reviewer_pWFp · 2025-08-03
> > **Rebuttal Reply**
> >
> > I thank the authors for their effective rebuttal. They have helpfully clarified my initial misunderstanding regarding the comparison to BayesRays (which was based on their training-phase results), and I am now convinced their final post-cleaning pipeline achieves comparable performance. To strengthen the paper and prevent similar confusion, I recommend including the full experimental results with all metrics in the main paper.
> >
> > However, my primary reservation persists regarding the uniform sampling strategy. I am concerned it may be overly aggressive and degrade fine geometric details in a way that photometric metrics may not capture. The paper would be significantly strengthened by including quantitative evidence to address this—such as the Chamfer Distance—to validate that surface integrity is preserved. I would welcome the thoughts of the other reviewers on this matter.

---

> > > ### Author Response · Authors · 2025-08-04
> > >
> > > We thank the reviewers for their helpful reviews and discussion, which help us improve the paper for future readers. Our focus was on measuring photometric degradation as conventionally done in NeRF cleanup papers, but **we agree with the reviewers that geometric evaluation should also be considered**.
> > >
> > > **TL;DR:** We conducted the three experiments suggested by the reviewers (Chamfer Distance, Depth consistency and training images PSNR) and our method achieves **comparable or better geometric and photometric preservation** compared to BayesRays, while being **2.5× faster in inference**.
> > >
> > > **Coverage ~92%-93%:**
> > >
> > > |Method|Chamfer Distance ↓|Depth (RMSE) ↓|Depth (MAE) ↓|PSNR ↑|
> > > |--------|------------------|--------------|-------------|------|
> > > | Ours ($\\lambda=0.01$)|**0.011**|**3.80**|**1.02**|**28.64**|
> > > | BayesRays (T=0.9)|**0.011**|3.91|1.07|28.51|
> > >
> > > **Coverage ~89%-90%:**
> > >
> > > |Method|Chamfer Distance ↓|Depth (RMSE) ↓|Depth (MAE) ↓|PSNR ↑|
> > > |--------|------------------|--------------|-------------|------|
> > > | Ours ($\\lambda=0.1$)|**0.011**|**3.96**|**1.04**|**28.50**|
> > > | BayesRays (T=0.7)|**0.011**|4.11|1.14|28.19|
> > >
> > > Importantly, these experiments strengthen our claim that our method is robust to the hyperparameter $\\lambda$, as choosing two different $\\lambda$ values resulted in only minor differences in geometric preservation.
> > >
> > > **Chamfer Distance:** Following reviewer pWFp03's suggestion, we measured Chamfer Distance between pre-cleanup and post-cleanup scenes. For each scene, we generated 35k-point point clouds using training cameras and NeRF reconstructions (pre- and post-cleanup), then computed Chamfer Distance between them.
> > >
> > > **Depth Consistency:** As suggested by reviewer fGuQ, to evaluate surface preservation using depth maps to check whether they remain consistent after cleanup on the training images. For the most accurate measurement, we compared rendered depth maps to **pseudo-gt depth maps** derived from a NeRF trained on both training and test images.
> > >
> > > **PSNR:** Following reviewer fGuQ's suggestion, we measured the PSNR of **training rendered images** after cleanup. We agree with the reviewer that validating scene integrity using training images is the correct approach, as this ensures that the core mechanism of NeRF reconstruction is preserved even through the cleanup phase.
> > >
> > > Both **geometric** (Chamfer Distance, depth consistency) and **photometric** (PSNR) evaluations confirm that our uniform sampling strategy preserves geometric details effectively. We thank the reviewers for their insights and the opportunity to strengthen our paper by adding these experimental results.
> > >
> > > Here are the full results of the above experiments:
> > >
> > > **Chamfer Distance:**
> > >
> > > | Method|Average|aloe|art|car|century|flowers|garbage|picnic|pikachu|pipe|plant|roses|table|
> > > |--------|---------|------|-----|-----|---------|---------|---------|--------|---------|------|-------|-------|-------|
> > > | Ours 0.01|0.011|0.015|0.008|0.011|0.009|0.013|0.005|0.010|0.008|0.012|0.012|0.015|0.014|
> > > | BayesRays 0.9|0.011|0.015|0.008|0.011|0.009|0.013|0.005|0.010|0.008|0.012|0.012|0.015|0.014|
> > > | Ours 0.1|0.011|0.015|0.008|0.011|0.009|0.013|0.004|0.010|0.008|0.012|0.012|0.016|0.014|
> > > | BayesRays 0.7|0.011|0.015|0.009|0.010|0.009|0.013|0.004|0.010|0.008|0.012|0.012|0.015|0.014|
> > >
> > > **PSNR:**
> > >
> > > | Method|Average|aloe|art|car|century|flowers|garbage|picnic|pikachu|pipe|plant|roses|table|
> > > |--------|---------|------|-----|-----|---------|---------|---------|--------|---------|------|-------|-------|-------|
> > > | Ours 0.01|28.64|25.66|29.55|22.88|28.79|26.92|25.23|28.74|34.96|28.23|30.14|30.43|32.16|
> > > | BayesRays 0.9|28.51|25.82|27.79|22.85|28.80|26.97|25.12|28.70|34.87|28.29|30.33|30.50|32.10|
> > > | Ours 0.1|28.50|25.71|29.41|22.82|28.69|26.84|25.09|28.62|34.72|28.12|30.08|30.00|31.96|
> > > | BayesRays 0.7|28.19|25.82|25.83|22.88|28.79|26.97|25.12|28.69|34.82|28.27|30.12|29.01|31.98|
> > >
> > > **Depth (RMSE):**
> > >
> > > | Method|Average|aloe|art|car|century|flowers|garbage|picnic|pikachu|pipe|plant|roses|table|
> > > |--------|---------|------|-----|-----|---------|---------|---------|--------|---------|------|-------|-------|-------|
> > > | Ours 0.01|3.80|0.38|16.15|3.81|1.97|0.90|1.72|1.54|0.69|12.50|3.27|1.77|0.89|
> > > | BayesRays 0.9|3.91|0.38|17.64|3.80|1.96|0.90|1.71|1.54|0.69|12.40|3.26|1.76|0.88|
> > > | Ours 0.1|3.96|0.39|16.44|3.85|2.21|0.91|1.76|1.57|0.70|13.14|3.29|1.78|1.46|
> > > | BayesRays 0.7|4.11|0.46|19.43|3.81|1.97|0.91|1.72|1.55|0.72|12.47|3.27|2.15|0.94|
> > >
> > > **Depth (MAE):**
> > >
> > > | Method|Average|aloe|art|car|century|flowers|garbage|picnic|pikachu|pipe|plant|roses|table |
> > > |--------|---------|------|-----|-----|---------|---------|---------|--------|---------|------|-------|-------|-------|
> > > | Ours 0.01|1.02 |0.10|4.16|1.13|0.50|0.36|0.47|0.32|0.23|2.64|1.40|0.56|0.43|
> > > | BayesRays 0.9 |1.07|0.10|4.70|1.13|0.50|0.35|0.47|0.32|0.23|2.62|1.40|0.56|0.43|
> > > | Ours 0.1|1.04 |0.10|4.18|1.13|0.50|0.36|0.47|0.33|0.23|2.76|1.41|0.56|0.44|
> > > | BayesRays 0.7 |1.14|0.10|5.50|1.13|0.50|0.35|0.47|0.32|0.23|2.64|1.40|0.59|0.43|

---

> > > > ### Comment · Reviewer_pWFp · 2025-08-05
> > > > **Rebuttal Reply**
> > > >
> > > > Thank authors for the updated experiments and the productive discussion with reviewers `fGuQ` and `XP5a`. While my initial suggestion was for a new benchmark on a high-quality GT dataset like DTU, I agree that the experiments proposed by the other reviewers can also address this concern.
> > > >
> > > > The proposed method is simple yet demonstrably effective, as supported by the new results. My initial concern regarding potential damage to fine structures has been partially alleviated; the updated experiments show that the photometric loss dominates in near-surface regions, preserving fine geometry.
> > > >
> > > > I appreciate the method's simplicity. Its potential to remove floaters without harming surface reconstruction would be highly impactful for the community. Given the paper's potential impact, its simplicity, and the extended rebuttal time, I suggest one final supplementary experiment to solidify the contribution.
> > > >
> > > > Run the proposed method and BayesRay on the mip-NeRF 360 using **dense view inputs**, demonstrating that it has a negligible influence when views are already sufficient for high-quality results.
> > > >
> > > > Should this experiment yield a promising result, I am prepared to raise my score to be positive.

---

> > > > > ### Author Response · Authors · 2025-08-08
> > > > > **mip-NeRF 360 using dense view inputs**
> > > > >
> > > > > We thank the reviewer for the constructive discussion and for suggesting this important experiment. We conducted the requested ablation study on the mip-NeRF 360 dataset with dense view inputs and found that **our method has a negligible influence** and it maintains high-quality results while being 2.5× faster in inference compared to BayesRays.
> > > > >
> > > > > **Results on mip-NeRF 360 dataset (average across all scenes):**
> > > > > | Method | PSNR ↑ | SSIM ↑ | LPIPS ↓ |
> > > > > |--------|--------|--------|---------|
> > > > > | Base Nerfacto | 27.74 | 0.789 | 0.181 |
> > > > > | Ours | 27.70 | 0.788 | 0.183 |
> > > > > | BayesRays | 27.73 | 0.789 | 0.181 |
> > > > >
> > > > > **Experimental setup:** We applied cleanup methods to Nerfacto trained for 30k iterations using dense training images following the Nerfstudio mip-NeRF 360 evaluation protocol as published in SIGGRAPH 2023, with the conventional train/test partition of every 8th image as test. We compared our method with $\\lambda=0.01$ to BayesRays with $T=0.9$.
> > > > >
> > > > > **Key findings:** Our method achieves reconstruction quality comparable to the baseline when views are sufficient, while delivering performance on par with BayesRays through a simpler, faster, and more efficient approach.
> > > > >
> > > > > The full results:
> > > > > | Method | Metric | Average | bicycle | bonsai | counter | garden | kitchen | room | stump |
> > > > > |--------|--------|---------|---------|--------|---------|--------|---------|------|-------|
> > > > > | Base | PSNR | 27.74 | 23.57 | 31.02 | 27.04 | 25.74 | 30.29 | 30.95 | 25.57 |
> > > > > | Ours | PSNR | 27.70 | 23.53 | 30.88 | 26.95 | 25.74 | 30.36 | 30.87 | 25.59 |
> > > > > | BayesRays | PSNR | 27.73 | 23.56 | 30.99 | 27.02 | 25.74 | 30.35 | 30.91 | 25.57 |
> > > > > | Base | SSIM | 0.789 | 0.557 | 0.927 | 0.834 | 0.731 | 0.900 | 0.903 | 0.671 |
> > > > > | Ours | SSIM | 0.788 | 0.556 | 0.924 | 0.832 | 0.732 | 0.899 | 0.902 | 0.670 |
> > > > > | BayesRays | SSIM | 0.789 | 0.557 | 0.926 | 0.833 | 0.731 | 0.900 | 0.903 | 0.671 |
> > > > > | Base | LPIPS | 0.181 | 0.400 | 0.066 | 0.146 | 0.212 | 0.077 | 0.106 | 0.262 |
> > > > > | Ours | LPIPS | 0.183 | 0.395 | 0.070 | 0.148 | 0.213 | 0.079 | 0.112 | 0.261 |
> > > > > | BayesRays | LPIPS | 0.181 | 0.400 | 0.066 | 0.146 | 0.212 | 0.077 | 0.107 | 0.262 |

---

> ### Comment · Reviewer_fGuQ · 2025-08-03
>
> I agree with reviewer pWFp that the degradation of fine geometric details might be a concern, and it would be cool to asses this quantitavely. Using Chamfer Distance on point clouds might, thought, be misleading, since there is no guarantee that the starting point cloud is correct, especially given the presence of floaters and other artifacts. I'm also not aware, given my lack of experise on the topic, of whether Chamfer Distance can be used to evaluate surfaces. In general, checking geometry in 3D in a quantitative way is difficult without a proper ground truth, particularly due to the poor representation of the initial NeRF geometry before any cleanup. There might be a significant difference just because artifacts were removed.
>
> A possible experiment could be to compare the PSNR between the training rendered images before and after the cleanup. Alternatively, it might be useful to check whether the surface normals or depth maps remain consistent before and after the cleanup, at least on the training images.
>
> Please correct me if I'm saying something wrong, or if this doesn't make sense in light of what the authors and reviewers think.

---

> > ### Comment · Reviewer_XP5a · 2025-08-04
> >
> > I also agree with reviewer pWFp. Verifying the pre and post resulting geometry after the cleanup process isn't degraded is an important experiment given the regularization is on the density.

---

> ### Comment · Reviewer_pWFp · 2025-08-08
>
> Thank authors again for the extended experiments. The dense-view experiment shows that the performance of free-space loss may slightly influence the performance, but marginally. Overall, I think authors demonstrate the free-space loss will not significantly degrade of original quality. I appreciate the simplicity of this method and will raise my score accordingly.

---

### Official Review · Reviewer_fGuQ · 2025-06-20

**Clarity:** 4
**Significance:** 3
**Originality:** 1
**Rating:** 4
**Confidence:** 3

**Summary:**

The paper focuses on removing unwanted floaters in NeRF, which appear in areas not detected during training. The authors propose a simple yet very effective way to supervise them using a Free Space Prior, if a region is not supposed to have any density, it should be zero.
They are faster and less costly compared to current SOTA methods.

**Questions:**

I don’t have any questions, my only concern is that I would like to see the method rival BayesRays in terms of PSNR while winning in coverage areas. I say this again because I imagine the current results are due to the Free Space Prior loss affecting all 3d points in the same way.

I’m really undecided between a 3 and a 4, mainly because of the limited novelty. I’m giving it a 4 just because the results show that it works really well, and I really liked the HTML visualization video used to show that.

**Ethical Concerns:**

["NO or VERY MINOR ethics concerns only"]

**Final Justification:**

Before, I said I was undecided between a 3 and a 4. Now, given all the new experiments and the hard work provided in the rebuttal, I am confident in my 4, now leaning more towards a 4+ than not.

**Limitations:**

Yes

**Quality:**

3

**Strengths And Weaknesses:**

The method shows great qualitative results; I really like the video HTML visualization.
The formulation is simple, makes sense, and it works.
The experiments, both quantitative and qualitative, are well done.

The only weakness I would mention is the novelty, since the contribution is only a loss applied post hoc (but since the post hoc step is fast, it’s not a problem).

I personally don’t like the fact that no reasoning is provided over the points that could be detected as “correct” and left untouched. This could strengthen the method and help it achieve better photometric results. Since the novelty is limited, there was room to explore this.
This would help avoid having the Free Space Prior loss negatively affect regions that are already well reconstructed.

---

> ### Author Rebuttal · Authors · 2025-07-29
>
> We thank the reviewer for the positive assessment and particularly appreciate the recognition of our visualization and experimental quality. Below are our clarifications and answers:
>
> ## Addressing Novelty Concerns
>
> While our core contribution is indeed a simple post-hoc loss function, we believe this simplicity is a **strength rather than a limitation**. The method's elegance lies in achieving SOTA results through a principled, theoretically grounded approach that generalizes across diverse NeRF architectures and datasets without requiring architectural modifications or learned priors.
>
> ## Selective Point Treatment
>
> The reviewer's suggestion about reasoning over "correct" points is insightful and represents an interesting direction for future work. While our current approach takes a simpler strategy, our **"Balancing Cleanup and Scene Integrity"** mechanism (line 157) naturally achieves similar outcomes through the interaction between losses. This creates three distinct regions of interaction between the losses:
>
> 1. **Unseen regions**: Only FSP loss acts, pushing densities to zero
> 2. **Empty seen regions**: Both losses agree on zero density
> 3. **Surface regions**: NeRF reconstruction rays provide strong, consistent supervision by repeatedly sampling the same locations, while the sparse FSP loss only occasionally samples these areas and therefore cannot overpower the reconstruction signal
>
> You can observe Figure 4 to see how density in empty regions is cleaned while preserving surface structure through this natural partitioning that allows our method to preserve well-reconstructed regions while cleaning artifacts.

---

### Official Review · Reviewer_XP5a · 2025-06-24

**Clarity:** 2
**Significance:** 2
**Originality:** 2
**Rating:** 4
**Confidence:** 2

**Summary:**

This work focuses on the problem of cleaning floaters from NeRF frameworks post single scene optimization. The proposed method introduces a post training procedure which extend the standard NeRF photometric loss with a empty space constraint regularization term that assumes the density should be zero everywhere. The effect intended affect this loss has is to ensure that the density in free space goes to zero, and to preserve existing information about the scene via ray sampling. Experiments show the approach to post clean state-of-the-art NeRF methods, as well as comparison with other works that solve the same problem.

**Questions:**

- Would be good if authors can mention if lambda was hand-chosen for different scenes. Given the proposed method and diversity in scenes, a best choice of lambda per scene would almost be expected.

**Ethical Concerns:**

["NO or VERY MINOR ethics concerns only"]

**Final Justification:**

My main concern was regarding the methodology and the core assumptions made and how those affect the generality and performance in different conditions. Reading the other reviews and discussions it looks similar concerns were raised.
During the rebuttal period the authors showcased further experiments that highlighted the robustness of the approach in different conditions while preserving/improving geometry and fidelity of base results.
I encourage the authors to improve on the manuscript, specifically the experiment section to address such concerns.

**Limitations:**

See weaknesses

**Quality:**

2

**Strengths And Weaknesses:**

Strength:
- Runtime for the proposed approach is far superior than competing baselines. Making it applicable and a good
- Method is simple and intuitive.

Weaknesses:
- The assumption that "there is nothing in unseen regions" is quite strong and may not hold in many real scenarios. Given that this assumption might break in non-object centric scenes, it would be better if explored properly.
- The regularization term mathematically, has adverse affects with the scene rays which have actual density. This may degrade the results on the actual object as seen by the PSNR results in Figure 5.
- Limited novelty.

Overall, the method could be useful to the community provided good trade-offs between speed and performance shown. It is however important to be more thorough with the experiments given the strong assumption this regularizer applies.

---

> ### Author Rebuttal · Authors · 2025-07-29
>
> We thank the reviewer for the feedback, we encourage the reviewer to view the **HTML visualization video** in the supplementary as one of the reviewers mentioned it as a helpful tool to understand the good results. Below are our clarifications and answers:
>
> ## "Nothing in Unseen Regions" Assumption
>
> This assumption is not strong, but rather **fundamental to NeRF reconstruction**. If training images do not cover an area, **it cannot be meaningfully reconstructed** - this is a core limitation of photometric optimization. Our contribution is in enforcing zero density in these regions instead of allowing arbitrary noise accumulation.
>
> Regarding the concern about "non-object centric scenes", we would like to highlight that our experimental evidence demonstrates effectiveness across **many non-object-centric scenes** throughout our paper. Notably, the **Nerfbusters dataset itself is not object-centric**. Our method performs well on complex indoor and outdoor scenes with multiple objects, backgrounds, and varied scene compositions, which we believe addresses the concern about applicability to non-object-centric scenarios.
>
> Additionally, our approach generalizes to many NeRF variants and scenarios such as underwater, semantic, and language-embedded NeRFs as shown in Figure 1, which distinguishes our method from other state-of-the-art cleanup approaches.
>
> ## The Regularization Term
>
> We believe there may be a misreading of Figure 5: **increasing $\\lambda$ actually improves PSNR** (while decreasing coverage as expected). This relationship can be verified in Table 2 of the supplementary material.
>
> It is true that the regularization term interacts with scene rays, but we discuss this extensively in the paper. **This is not a limitation, but the core mechanism of our method**.
>
> Our "**Balancing Cleanup and Scene Integrity**" section (line 157) explains the 3 regions of interaction between the losses:
> - **Unseen regions**: Only FSP loss acts, pushing densities to zero
> - **Empty seen regions**: Both losses agree on zero density
> - **Surface regions**: NeRF reconstruction rays provide strong, consistent supervision by repeatedly sampling the same locations, while the sparse FSP loss only occasionally samples these areas and therefore cannot overpower the reconstruction signal
>
> These are illustrated in Figure 4, which shows how density in empty regions is cleaned while preserving surface structure.
>
> ## Universal $\\lambda$ Parameter
>
> **$\\lambda$ is not hand-chosen for different scenes** as mentioned in line 178. **All visualizations in the paper were generated with $\\lambda=0.1$** across different NeRF methods and scene characteristics, demonstrating universal applicability.
>
> Our extensive ablation study (Appendix A.5) shows that our method's mechanism is more important than precise $\\lambda$ tuning:
>
> > *"Our method is robust to a wide range of $\\lambda$ values as can be seen in Table 2, $\\lambda \\in [10^{-5}, 1]$ yields both high PSNR (17.35–17.69) and high coverage (87.57%–96.45%). This robustness stems from our method's design, which optimizes regions where the reconstruction loss is inactive."*

---

> > ### Comment · Reviewer_XP5a · 2025-08-05
> > **Concerns remain around generalization**
> >
> > I thank the authors for clarifying the robustness around parameter \lambda. And also for taking the time to answer and provide additional results regarding depth reconstruction. This highlights that geometry is not compromised post training clean-up.
> > Regarding the base assumption behind the method. I am still skeptic about a regularization term that zeros out the density everywhere. One of the core results of NeRFs is their ability to interpolate over a subset of frames and provide a smooth rendering trajectory of the scene. This interpolation capability is a core property of coordinate networks which effectively fills in the blanks. My concerns remains around performance degradation when frame sample is sparse, which in this case can effect results once post-trained.

---

> > > ### Author Response · Authors · 2025-08-08
> > > **Sparse view performance**
> > >
> > > We thank the reviewer for the constructive discussion and are pleased that our clarifications addressed concerns regarding the λ hyperparameter and geometry preservation. We particularly appreciate the important point raised about sparse view performance, as this tests a fundamental capability of NeRFs.
> > >
> > > Regarding the concerns around performance degradation when the frame sample is sparse, we conducted an ablation study for sparse views on the Garden scene from mipNeRF-360 dataset. These results demonstrate that our method not only preserves NeRF's interpolation capabilities under sparse training conditions but actually **improves perceptual quality (SSIM/LPIPS) while maintaining comparable PSNR**. This addresses the reviewer's concern about our regularization approach compromising fundamental NeRF properties.
> > >
> > >
> > > 24 training images (Every 8th image):
> > > | Method | PSNR ↑ | SSIM ↑ | LPIPS ↓ |
> > > |--------|--------|--------|---------|
> > > | Base | 22.32 | 0.637 | 0.261 |
> > > | Ours | 22.31 | **0.640** | **0.257** |
> > > | BayesRays | **22.33** | 0.637 | 0.259 |
> > >
> > > 47 training images (Every 4th image):
> > > | Method | PSNR ↑ | SSIM ↑ | LPIPS ↓ |
> > > |--------|--------|--------|---------|
> > > | Base | 24.41 | 0.703 | 0.226 |
> > > | Ours | 24.40 | **0.705** | **0.222** |
> > > | BayesRays | **24.42** | 0.703 | 0.226 |
> > >
> > > 93 training images (Every 2nd image):
> > > | Method | PSNR ↑ | SSIM ↑ | LPIPS ↓ |
> > > |--------|--------|--------|---------|
> > > | Base | 25.33 | 0.724 | 0.212 |
> > > | Ours | 25.35 | **0.727** | **0.211** |
> > > | BayesRays | **25.38** | 0.725 | 0.212 |
> > >
> > > 161 training images (Regular experiment):
> > > | Method | PSNR ↑ | SSIM ↑ | LPIPS ↓ |
> > > |--------|--------|--------|---------|
> > > | Base | 25.74 | 0.731 | 0.212 |
> > > | Ours | **25.74** | **0.732** | 0.213 |
> > > | BayesRays | **25.74** | 0.731 | **0.212** |
> > >
> > > **Experimental setup:** Instead of using most of the images for training and only every 8th image as a test image, we swapped the rolls and used most of the images as test and only few as training images (every 8th, 4th or 2nd image in three different experiments). We applied cleanup methods to Nerfacto trained for 30k iterations using sparse training images, comparing our method with $\\lambda=0.01$ and BayesRays with $T=0.9$.

---

### Official Review · Reviewer_yNLy · 2025-06-24

**Clarity:** 2
**Significance:** 3
**Originality:** 3
**Rating:** 4
**Confidence:** 4

**Summary:**

The authors propose a method to reduce the density of sampled points in 3D space, thereby eliminating artifacts and improving the quality of novel view synthesis in NeRF.

**Questions:**

See Weaknesses.

**Ethical Concerns:**

["NO or VERY MINOR ethics concerns only"]

**Final Justification:**

After reviewing all reviewers’ comments and the authors’ responses, I believe all of my concerns have been addressed, and I will raise my score to 4.

**Limitations:**

Yes.

**Paper Formatting Concerns:**

No.

**Quality:**

3

**Strengths And Weaknesses:**

**Strengths:**

The authors propose a simple and effective method that reduces the density of sampled points in 3D space to eliminate artifacts.

**Weaknesses:**

1. It should be clarified whether the training time reported in Table 1 refers to training from scratch or fine-tuning. If it corresponds to fine-tuning, the number of fine-tuning iterations should be specified, and an ablation study on the number of fine-tuning iterations should be provided.
2. The authors emphasize that the proposed method has faster inference speed, but there are no optimizations to the inference process presented in the paper. I am puzzled as to how the authors arrived at this conclusion.
3. Is the hyperparameter λ related to the number of images in a scene? For example, if one scene contains 10 images and another contains 100, the sampling (or optimization) frequency would differ across scenes. However, the *FSP* loss reduces the density in space at a fixed frequency, which could lead to different trade-offs in different scenes.
4. I believe the number of sampling points is also an important hyperparameter, but the paper lacks an ablation study on this aspect.
5. The organization of the experimental section is not clear enough. Some important experimental details should be included in the main text, such as the ablation study of the hyperparameter λ and the comparison between fine-tuning and training from scratch.

---

> ### Author Rebuttal · Authors · 2025-07-29
>
> We thank the reviewer for the feedback, we encourage the reviewer to view the **HTML visualization video** in the supplementary as one of the reviewers mentioned it as a helpful tool to understand the good results. Below are our clarifications and answers:
>
> ## Training Time Clarification
>
> The training time refers to the "**cleanup phase**" and we will clarify this in the paper. **Each method uses a fundamentally different training approach**: our method performs fine-tuning using the Free Space Prior and Reconstruction Losses, Nerfbusters fine-tunes with a large pre-trained 3D diffusion model, while BayesRays collects Hessian data for uncertainty estimation. The details about each method's distinct training phase are in lines 274-277:
>
> > *"For training time, Nerfbusters employs a pre-trained 3D diffusion model to fine-tune NeRF per scene, resulting in a high training time of approximately 20 minutes per scene. BayesRays is faster, taking 37 seconds to gather data for Hessian computation, while our approach is the fastest, completing cleanup training in just 25 seconds."*
>
> As for the number of iterations and samples, it is stated in the implementation details (Lines 455-456), we will clarify that in the runtime section too:
>
> > *"In our method, density optimization was achieved with 1,000 iterations using $2^{17}$ randomly sampled points across the 3D scene space."*
>
> ## Inference Speed Advantage
>
> The reviewer asked how we achieve faster inference without optimizations to the inference process. The key difference lies in the **inference architecture**: BayesRays requires additional uncertainty thresholding computations during rendering, resulting in 506ms per frame, while our method uses only the fine-tuned NeRF without any additional overhead, achieving 205ms per frame on the same hardware - 2.5× faster than BayesRays, which is the only SOTA method achieving comparable cleanup quality.
>
> ## Hyperparameter $\\lambda$ Robustness
>
> We understand your concern regarding different scenes with different numbers of images. We claim that the method mechanism is more important than the specific $\\lambda$ parameter. To clarify, **all visualizations in the paper were generated with $\\lambda=0.1$** across different NeRF methods and scene characteristics, as mentioned in line 178.
>
> This robustness can be explained by two key mechanisms:
>
> 1. **Fixed sampling strategy**: The number of samples and number of rays are fixed in each iteration, regardless of the number of images in the scene. This ensures consistent optimization behavior across different scene configurations.
>
> 2. **Balanced loss interaction**: Our "Balancing Cleanup and Scene Integrity" section explains the 3 regions of interaction between the losses:
>    - **Unseen regions**: Only FSP loss acts, pushing densities to zero
>    - **Empty seen regions**: Both losses agree on zero density
>    - **Surface regions**: NeRF reconstruction rays provide strong, consistent supervision by repeatedly sampling the same locations, while the sparse FSP loss only occasionally samples these areas and therefore cannot overpower the reconstruction signal
>
> You can observe Figure 4 to see how density in empty regions is cleaned while preserving surface structure through this balanced interaction.
>
> Moreover, our ablation study demonstrates robust performance across a very large range of $\\lambda$ values (Appendix A.5 - Choosing $\\lambda$):
>
> > *"Our method is robust to a wide range of $\\lambda$ values as can be seen in Table 2, $\\lambda \\in [10^{-5}, 1]$ yields both high PSNR (17.35–17.69) and high coverage (87.57%–96.45%). This robustness stems from our method's design, which optimizes regions where the reconstruction loss is inactive."*
>
> ## The Organization of the Experimental Section
>
> Thank you for this feedback, which helps us improve the paper. We will move the ablation study of hyperparameter $\\lambda$ and the comparison between fine-tuning and training from scratch to the main text.

---

> > ### Comment · Reviewer_yNLy · 2025-08-05
> >
> > the authors have addressed most of my concerns; however, I would appreciate an explanation for the choice of the number of sampling points 𝑁 mentioned in line 149.

---

> > > ### Author Response · Authors · 2025-08-05
> > >
> > > We thank the reviewer for this suggestion. **We conducted an ablation study on hyperparameter $N$ as requested by the reviewer**.
> > >
> > > **Rationale for N selection:** $N$ should be proportional to the reconstruction loss sampling density. In each iteration, we use 4,096 rays with 352 samples per ray (\~1.45M samples) for reconstruction loss. We chose $N=2^{17}$ (\~131K samples), which is approximately one order of magnitude smaller, ensuring the FSP loss provides a meaningful cleanup signal while adding minimal computational overhead.
> > >
> > > **Ablation results on the Pikachu scene (λ=0.1):**
> > >
> > > $N=2^1$: 22.70s cleanup runtime, 24.88 PSNR - Minimal cleanup effect
> > >
> > > $N=2^{17}$: 25.23s runtime, 26.13 PSNR - Balanced quality-efficiency trade-off
> > >
> > > $N=2^{20}$: 39.55s runtime, 26.30 PSNR - Marginal quality improvement but doubled cleanup time
> > >
> > > **The ablation study confirms our rationale:** $N=2^{17}$ achieves substantial cleanup improvements with only \~10% additional runtime compared to minimal sampling, while avoiding the diminishing returns of higher $N$ values. There is a wide range of hyperparameter $N$ values that can be effectively used, depending on user preferences for the quality-efficiency trade-off.
> > >
> > > | $N$ | Cleanup Time (s) $\\downarrow$ | PSNR $\\uparrow$ | SSIM $\\uparrow$ | LPIPS $\\downarrow$ | Coverage $\\uparrow$ | Dice $\\uparrow$ |
> > > |-------------|------------------------|------------------|------------------|---------------------|---------------------|------------------|
> > > | $2^0$ | 22.70 | 24.88 | 0.87 | 0.08 | 88.77 | 0.87 |
> > > | $2^1$ | 23.62 | 25.28 | 0.88 | 0.08 | 89.67 | 0.87 |
> > > | $2^2$ | 23.30 | 25.46 | 0.88 | 0.08 | 89.62 | 0.87 |
> > > | $2^3$ | 23.28 | 25.42 | 0.87 | 0.09 | 90.18 | 0.88 |
> > > | $2^4$ | 23.37 | 25.63 | 0.87 | 0.09 | 90.28 | 0.88 |
> > > | $2^5$ | 23.37 | 25.59 | 0.87 | 0.09 | 90.78 | 0.89 |
> > > | $2^6$ | 23.28 | 25.55 | 0.87 | 0.09 | 90.22 | 0.88 |
> > > | $2^7$ | 23.29 | 25.76 | 0.88 | 0.07 | 88.23 | 0.87 |
> > > | $2^8$ | 23.41 | 25.64 | 0.88 | 0.07 | 88.42 | 0.88 |
> > > | $2^9$ | 23.21 | 25.66 | 0.88 | 0.08 | 89.31 | 0.88 |
> > > | $2^{10}$ | 23.29 | 25.65 | 0.88 | 0.08 | 88.60 | 0.88 |
> > > | $2^{11}$ | 23.40 | 25.94 | 0.88 | 0.08 | 88.43 | 0.88 |
> > > | $2^{12}$ | 23.51 | 26.01 | 0.88 | 0.08 | 88.57 | 0.88 |
> > > | $2^{13}$ | 23.88 | 25.97 | 0.88 | 0.08 | 89.12 | 0.88 |
> > > | $2^{14}$ | 23.88 | 25.98 | 0.88 | 0.08 | 88.29 | 0.88 |
> > > | $2^{15}$ | 24.11 | 26.11 | 0.89 | 0.07 | 87.89 | 0.88 |
> > > | $2^{16}$ | 24.58 | 26.07 | 0.89 | 0.07 | 87.70 | 0.88 |
> > > | $2^{17}$ | 25.23 | 26.13 | 0.89 | 0.07 | 87.85 | 0.88 |
> > > | $2^{18}$ | 27.15 | 26.16 | 0.89 | 0.07 | 87.22 | 0.87 |
> > > | $2^{19}$ | 31.36 | 26.21 | 0.89 | 0.07 | 86.62 | 0.87 |
> > > | $2^{20}$ | 39.55 | 26.30 | 0.89 | 0.07 | 86.97 | 0.88 |
> > > | $2^{21}$ | 56.46 | 26.23 | 0.89 | 0.07 | 86.64 | 0.87 |
> > > | $2^{22}$ | 89.85 | 26.22 | 0.90 | 0.06 | 86.21 | 0.87 |

---

### Note · Authors · 2025-08-12

We thank the reviewers for a constructive and productive discussion.  We are glad to have addressed all major concerns through clarifications and additional experiments, which further strengthen the paper and demonstrate its robustness, efficiency, and practical impact.

---

### Decision · Program_Chairs · 2025-09-17

**Decision:**

Accept (poster)

**Comment:**

This paper introduces a fast, post-hoc cleanup method for NeRF that effectively removes floaters. The proposed method incorporates several new ideas and advantages, including the use of a Maximum-a-Posteriori (MAP) approach with a simple global prior, explicit optimization in unseen regions, and improved efficiency in both computation and memory usage.

The paper received consistent final ratings of "borderline accept" from all reviewers..

The initial reviews were mostly negative, as the reviewers expressed concerns about generalizability, the core assumption of the method, and possible degradation of fine geometric details. In particular, the reviewers raised questions about how to avoid unintentionally changing parts of the scene due to the loss and requested to verify that the pre- and post-cleanup geometry does not degrade.

In response to these concerns, the authors conducted extensive experiments during their rebuttal and the subsequent post-rebuttal discussion. As a result, all reviewers developed a positive impression of the paper, appreciating its simplicity and effective results. After reviewing the comments and discussions, the area chair concurs with the reviewers' assessments and recommends accepting the paper.